# Subsurface ocean warming preceded Heinrich Events

Lars Max [1✉], Dirk Nürnberg [2], Cristiano M. Chiessi [3], Marlene M. Lenz [4] & Stefan Mulitza [1]

Although the global environmental impact of Laurentide Ice-Sheet destabilizations on glacial climate during Heinrich Events is well-documented, the mechanism driving these ice-sheet instabilities remains elusive. Here we report foraminifera-based subsurface (~150 m water depth) ocean temperature and salinity reconstructions from a sediment core collected in the western subpolar North Atlantic, showing a consistent pattern of rapid subsurface ocean warming preceding the transition into each Heinrich Event identified in the same core of the last 27,000 years. These results provide the first solid evidence for the massive accumulation of ocean heat near the critical depth to trigger melting of marine-terminating portions of the Laurentide Ice Sheet around Labrador Sea followed by Heinrich Events. The repeated build-up of a subsurface heat reservoir in the subpolar Atlantic closely corresponds to times of weakened Atlantic Meridional Overturning Circulation, indicating a precursor role of ocean circulation changes for initiating abrupt ice-sheet instabilities during Heinrich Events. We infer that a weaker ocean circulation in future may result in accelerated interior-ocean warming of the subpolar Atlantic, which could be critical for the stability of modern, marine-terminating Arctic glaciers and the freshwater budget of the North Atlantic.

[1] MARUM—Center for Marine Environmental Sciences, University of Bremen, Bremen 28359, Germany. [2] GEOMAR, Helmholtz Centre for Ocean Research Kiel, Kiel 24148, Germany. [3] School of Arts, Sciences and Humanities, University of São Paulo, São Paulo, Brazil. [4] Institute of Geology and Mineralogy, University of Cologne, Cologne 50674, Germany. ✉email: lmax@marum.de

The deposition of ice-rafted debris layers in the glacial North Atlantic (known as Heinrich Events) provides evidence for a substantial freshwater release via melting icebergs in response to past instabilities of the Laurentide Ice Sheet (LIS)[1–3]. Although the impact of Heinrich Events on global climate has been studied thoroughly during the last decades[4–6], the driving mechanism behind the episodic iceberg-discharge during Heinrich Events is a matter of ongoing debate[7–10]. A commonly discussed hypothesis is that Heinrich Events were initiated by the periodic unstable flow of the LIS, controlled by internal ice-sheet oscillations under otherwise stable environmental conditions, known as "binge-purge" hypothesis[7]. It is assumed that the massive freshwater release during Heinrich Events initiated strong disruptions of the Atlantic Meridional Overturning Circulation (AMOC)[3] and surface-ocean cooling of the North Atlantic[4]. However, surface ocean temperature and ice-rafted debris (IRD) proxy data from the high latitudes of the North Atlantic indicate that surface-ocean cooling occurred hundreds to thousands of years earlier than ice-rafting events in the North Atlantic[10]. Moreover, proxy data of AMOC strength show that deep ocean-circulation weakened prior to Heinrich Events in the North Atlantic[6]. Other studies propose a strong impact of weaker overturning circulation on LIS dynamics and the onset of Heinrich Events[9,11–13]. Based on numerical modelling simulations, a reduced AMOC would lead to strong subsurface warming and rapid retreat of ice-shelves around the Labrador Sea that is followed by the destabilization of the LIS during Heinrich Events[9,13]. Proxy data of bottom-water temperatures show that the mid-depth North Atlantic indeed warmed prior to Heinrich Events and thus seem to support this mechanism[14]. Nevertheless, proxy records reflecting subsurface ocean temperature variability near the Labrador Sea, close to the grounding line of marine-based portions of the LIS, are not available yet, hampering the evaluation of subsurface ocean warming as the trigger of Heinrich Events.

We studied fluctuations in subsurface ocean conditions relative to the occurrence of Heinrich Events at a site located to the east of Newfoundland in the subpolar western North Atlantic (marine sediment core GeoB18530-1; 42° 50′ N, 49° 14′ W, 1888 m water depth; Fig. 1a) at high temporal resolution (~250 years on average) over the last 27,000 years. We analysed the Mg/Ca ratio of subsurface dwelling planktonic foraminifera *Neogloboquadrina pachyderma* sinistral (*N. pachyderma* sin.) as a proxy for subsurface temperatures (subSST$_{Mg/Ca}$ at ~150 m water depth) from site GeoB18530-1 (see "Methods" and Supplementary Figs. 1–3). Combined information from subSST$_{Mg/Ca}$ data and the stable oxygen isotopic composition ($\delta^{18}$O) of *N. pachyderma* sin. allow calculating the regional ice-volume-corrected oxygen isotopic composition of seawater ($\delta^{18}$O$_{ivc-sw}$) as a proxy for salinity (see "Methods"). The chronostratigraphy of sediment core GeoB18530-1 is based on 20 accelerator mass spectrometer $^{14}$C ages spanning the last 35,000 years (Supplementary Fig. 4 and Supplementary Table 1). Site GeoB18530-1 is located at the southern boundary of the North Atlantic Subpolar Gyre (Fig. 1a). Today this location is under the influence of warm and saline waters of the North Atlantic Current (NAC), which is an integral part of the upper branch of the AMOC. Observational and modelling data show that modern inter-annual to decadal variability of temperature and salinity in the study area is controlled by Subpolar Gyre dynamics[15], suggesting that site GeoB18530-1 is a sensitive recorder of changes in ocean temperature and salinity. Sediment core GeoB18530-1 is ideally located to investigate the relative timing of past ocean dynamics in temperature and salinity against Heinrich Events associated with LIS instabilities within Hudson Strait because of: (i) its location close to the exit of the Labrador Sea within the IRD-belt[16], the North Atlantic gateway of the main iceberg trajectory associated with Heinrich Events (Fig. 1b); (ii) the well-defined IRD layers deposited during Heinrich Events (Fig. 2a); and (iii) the fact that all proxy records were established from the same

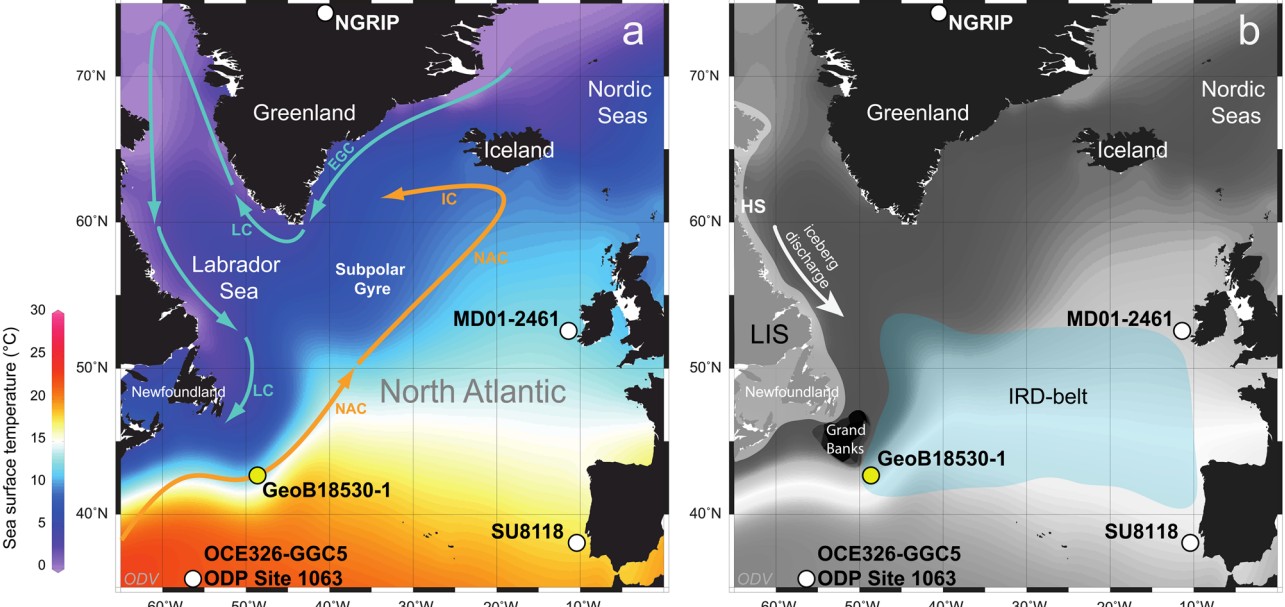

**Fig. 1 Modern surface-ocean conditions in the North Atlantic, the extension of the Laurentide Ice Sheet (LIS) and the IRD-belt in the North Atlantic during the Last Glacial Maximum. (a)** Annual mean sea surface temperature[20] (colour shading) and surface circulation[15] (arrows) in the study area. Yellow dot: location of core site GeoB18530-1 (42° 50′ N, 49° 14′ W; 1,888 m water depth; this study); white dots: location of reference core sites MD01-2461 (51°45′ N, 12° 55′ W; 1153 m water depth), SU8118 (37° 46′ N, 10°11′W; 3135 m water depth), OCE326-GGC5/ODP Site 1063 (33° 42′ N, 57° 35′ W; 4550 m water depth), and North Greenland Ice Core Project (NGRIP; 75° 5′ N, 42° 17′ W). EGC East Greenland Current, IC Irminger Current, LC Labrador Current, NAC North Atlantic Current. **(b)** Area shaded in white: LIS extent[53]; area shaded in green; IRD-belt in the North Atlantic[16]; HS Hudson Strait. This map was generated with Ocean Data View[54].

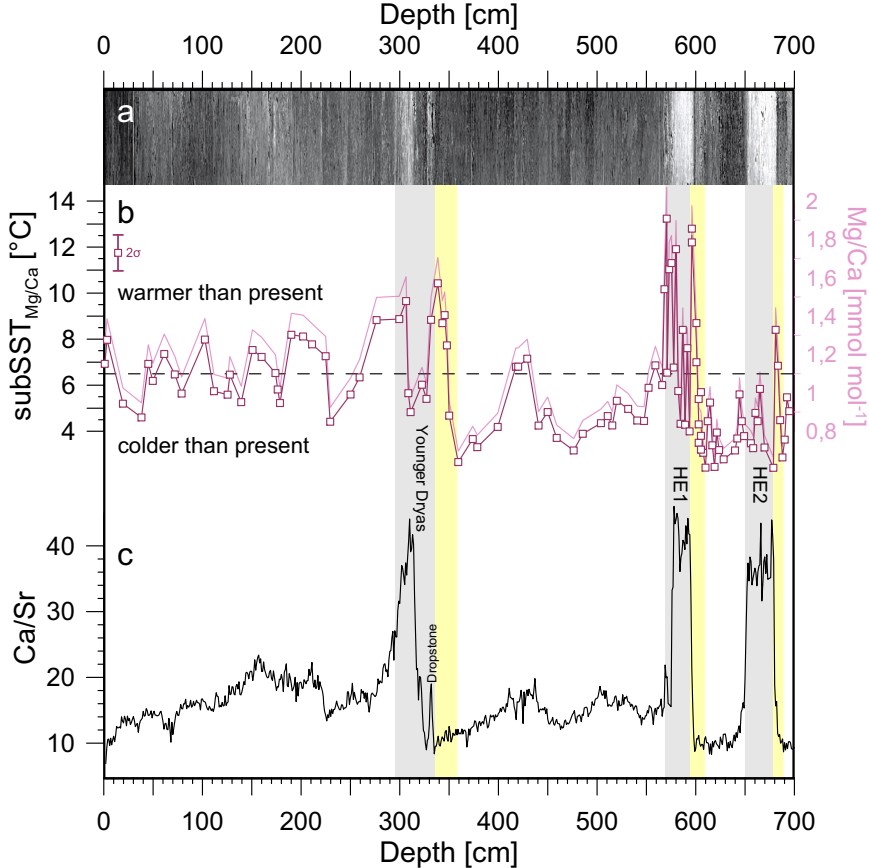

**Fig. 2 Proxy records from sediment core GeoB18530-1 versus core depth in comparison to digital core image. a** High-resolution digital core image[17]. **b** Foraminiferal Mg/Ca$_{N.\ pachyderma\ sin.}$ ratios and calculated subsurface temperatures (subSST$_{Mg/Ca}$) with analytical uncertainty (2σ). **c** X-ray fluorescence scanning-derived calcium to strontium ratios (Ca/Sr). yellow bars = phases of increases in subSST$_{Mg/Ca}$ into Heinrich Events and the Younger Dryas; grey bars = Heinrich Layers (HE2 = Heinrich Event 2, HE1 = Heinrich Event 1) and the Younger Dryas. Dashed horizontal line in Fig. 1b indicates modern temperature at ~150 m water depth close to site GeoB18530-1[20].

sediment core allowing robust determination of past ocean–ice-sheet interactions without temporal offsets and independent of any age modelling issues.

## Results

**Subsurface ocean warming prior to Heinrich Events.** The most intriguing finding in our records is a recurring pattern of massive rises of subSST$_{Mg/Ca}$ in the subpolar western North Atlantic (Fig. 2b). Another salient feature is the presence of very well-preserved IRD layers at site GeoB18530-1[17] (Fig. 2a). To further characterize the IRD layers we used the elemental ratio of calcium to strontium (Ca/Sr) from bulk sediments of core GeoB18530-1 as a proxy for detrital carbonate associated with Heinrich Event 1 and 2 [2,18,19] (see "Methods"). The IRD layers are characterized by elevated Ca/Sr ratios as expected from the high content of detrital carbonate are originating from Paleozoic limestone and dolostone from Hudson Bay and Hudson Strait[2] (Fig. 2c). Remarkably, the onset of subsurface ocean warmings clearly precedes the deposition of Heinrich Layers by several centimetres with respect to core depth (Fig. 2). It is important to note that the timing of subSST$_{Mg/Ca}$ increases prior to Heinrich Events is systematic and independent of any age model uncertainties. A closer look reveals that warmest subsurface ocean temperatures appear synchronous to the onset of IRD deposition during Heinrich Events (Fig. 2b). The timing of the subSST$_{Mg/Ca}$ warming peaks is further constrained by radiocarbon ages, suggesting a calibrated median age of ~17.1 ka BP at the beginning of Heinrich Event 1 and a

calibrated median age of ~25.4 ka BP close to the warming peak prior to Heinrich Event 2 (Supplementary Table 1). Subsurface temperatures rise to 8.4 °C and 12.5 °C at the onset of IRD layers of Heinrich Events 2 and 1, respectively (Fig. 2b, c). Modern hydrographic data close to site GeoB18530-1[20] exhibit a subsurface ocean temperature of ~7 °C for the inferred habitat depth range of *N. pachyderma* sin. (see Supplementary Fig. 3).

After the onset of Heinrich Events, subsurface waters experienced a phase of rapid cooling and freshening (Fig. 3). At this point, the cooling and freshening signals describe the well-known response to massive meltwater intrusions in the subpolar North Atlantic during Heinrich Events[2]. A second rise in temperature and salinity is evident at the later phase of Heinrich Events, most pronounced during Heinrich Event 1 (Fig. 3). This is consistent with findings derived from Heinrich layers of the central North Atlantic showing that Heinrich Event 1 is subdivided into an early phase (Heinrich Event 1.1; 17.1–15.5 ka BP) and a late phase (Heinrich Event 1.2; 15.9–14.3 ka BP) of IRD deposition, interpreted as two different ice-stream advances[21]. In particular, the synchronous onset of marked IRD deposition when subSST$_{Mg/Ca}$ appears to be warmest provides strong evidence for a causal role of subsurface ocean temperatures in triggering Heinrich Events of the last 27,000 years (Fig. 3).

**Subsurface ocean warming linked to AMOC slowdowns.** Further comparison of proxy records of North Atlantic–North

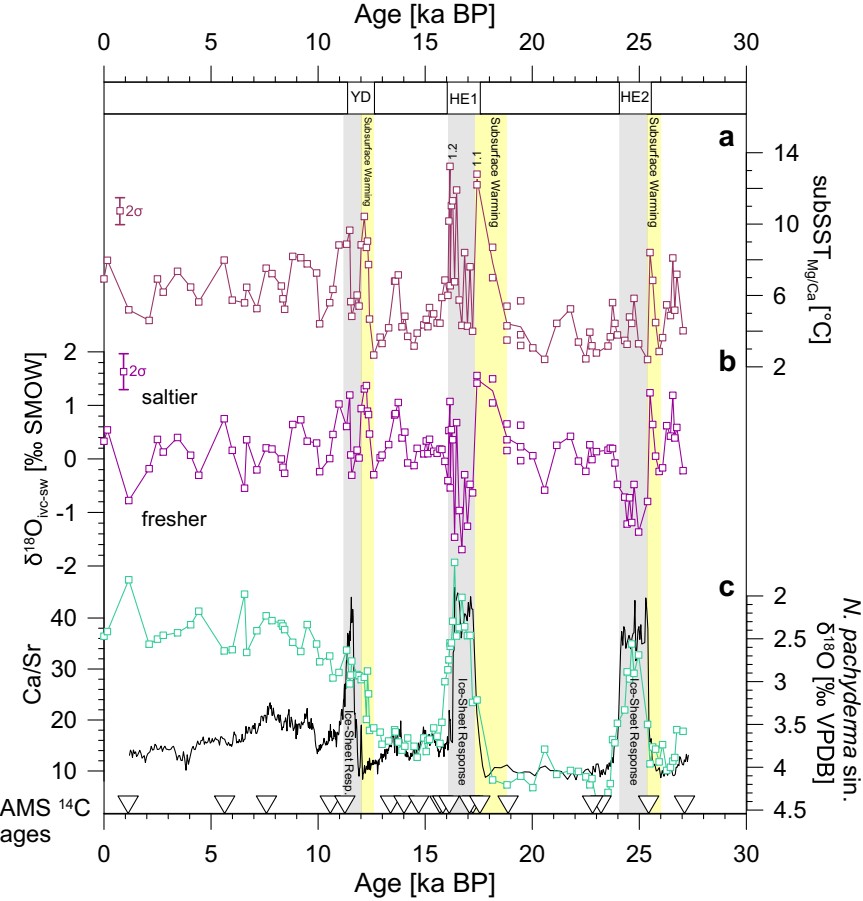

**Fig. 3 Subsurface ocean variability and Heinrich Events over the last 27,000 years from sediment core GeoB18530-1. a** Subsurface temperatures (subSST$_{Mg/Ca}$) with analytical uncertainty (2σ). **b** Ice-volume-corrected oxygen isotopic composition of seawater (δ$^{18}$O$_{ivc-sw}$) with analytical uncertainty (2σ). **c** Oxygen isotope composition (δ$^{18}$O) of *Neogloboquadrina pachyderma* sinistral and X-ray fluorescence scanning-derived element ratio of calcium to strontium (Ca/Sr). Bottom: Open triangles indicate calibrated radiocarbon ages (AMS $^{14}$C ages). yellow bars = phases of increases in subSST$_{Mg/Ca}$ into Heinrich Events and the Younger Dryas (YD); grey bars = Heinrich Layers (HE2 Heinrich Event 2, HE1 Heinrich Event 1 with Heinrich Event 1.1 and Heinrich Event 1.2 indicated)[21] and YD.

Greenland Ice Core Project (NGRIP) temperature variability and AMOC strength yields three key observations (Fig. 4). First, pronounced temperature increases prior to Heinrich Events are neither visible in the NGRIP[22,23] record of atmospheric temperatures, nor in North Atlantic sea surface temperatures[4] over the last 27,000 years (Fig. 4a, e). Moreover, North Atlantic sea surface temperatures cool (warm) when subSST$_{Mg/Ca}$ warm (cool). The opposing temperature trends are particularly evident prior to Heinrich Events as well as the Younger Dryas and we conclude that subsurface water masses must be very well isolated from the surface ocean of the subpolar Atlantic during these intervals (Fig. 4b, e).

Second, early subsurface ocean warming is also evident from site MD01-2461 located in the subpolar eastern North Atlantic, being most pronounced prior to Heinrich Event 1[24] (Fig. 4d). The correlation with elevated IRD flux at the same site is supposed to reflect a situation, in which anomalously warm conditions in the subpolar eastern North Atlantic caused ablation of the marine ice-margin of the British Ice Sheet[24]. This is in line with palaeoglaciological studies describing a significant ice-margin retreat of the British-Irish Ice Sheet between ~19 and 17 ka BP associated with catastrophic disintegration of the North Sea ice-bridge and an enormous outburst from a meltwater lake into the southern North Sea[25]. Based on the close correspondence of subSST$_{Mg/Ca}$ increases at both margins of the subpolar North

Atlantic (Fig. 1), we hypothesize that a large volume of heat was stored in the interior of the subpolar North Atlantic prior to Heinrich Event 1. Our interpretation is in line with previous findings from benthic Mg/Ca bottom water temperature reconstructions of the mid-depth Northwest Atlantic proposing significant warming of the interior-ocean prior to Heinrich Events[14].

Third, proxy data of AMOC variability[6,26] show that deep-ocean circulation weakened during Heinrich Stadial 2 and 1, preceding Heinrich Events by 1–2 kyrs[14] (Fig. 4f). The signal of an early decline in AMOC preceding Heinrich Events by 1–2 kyrs exceeding the typical range of age model uncertainties for paleo-reconstructions of several hundreds of years and seems to be quite robust. Our subSST$_{Mg/Ca}$ record indicates the build-up of ocean heat in the subpolar western North Atlantic during Heinrich Stadials and the Younger Dryas (Fig. 4b). Accordingly, we found calibrated mean ages for the beginning of subsurface ocean warming of ~25.9 ka BP during Heinrich Stadial 2, ~18.6 ka BP at the transition to Heinrich Stadial 1 and ~12.5 ka BP at the beginning of the Younger Dryas, during repeated slowdowns of the AMOC. These results point to a close temporal relationship between weaker overturning circulation and the increase in ocean heat content in the subpolar western North Atlantic (Fig. 4b, f). However, we found a delay in IRD deposition towards the end of the Younger Dryas interval at site GeoB18530-1 (Fig. 4b, c). This

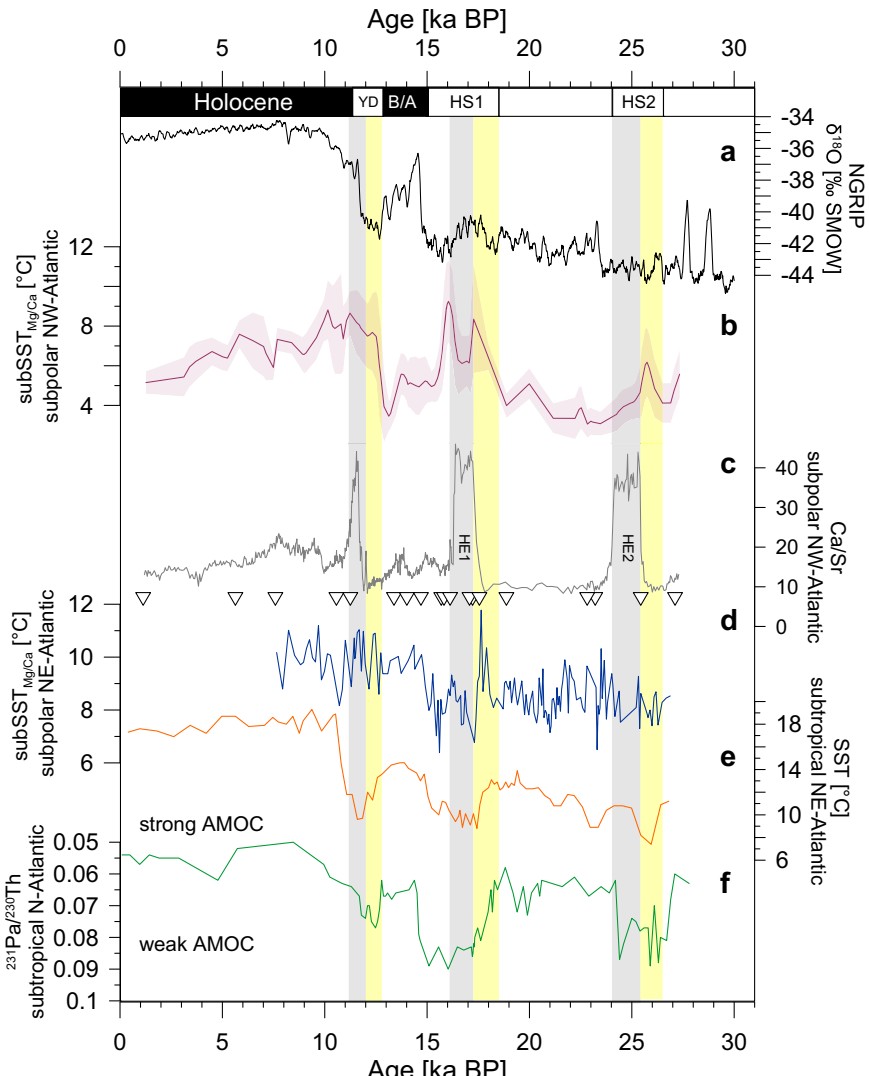

**Fig. 4 Comparison of proxy records from sediment core GeoB18530-1 to representative Greenland– North Atlantic proxy records over the last 27,000 years. a** North Greenland Ice Core Project (NGRIP)[22,23] stable oxygen isotopic composition ($\delta^{18}O$) reflecting atmospheric temperatures over Greenland. **b** Mean ages from age model ensemble for subsurface temperatures of core GeoB18530-1 with 95% confidence interval (this study). **c** X-ray fluorescence scanning-derived calcium to strontium ratios (Ca/Sr) (grey line) and age control points derived from calibrated AMS [14]C ages (open triangles) from core GeoB18530-1 (this study). **d** Subsurface temperatures from core MD01-2461 (eastern subpolar North Atlantic)[24]. **e** Alkenone sea surface temperatures from core SU8118 (eastern subtropical North Atlantic)[4]. **f** Pa/Th ratio reflecting the strength of the Atlantic Meridional Overturning Circulation (AMOC). Time series from 0 to 19 ka BP derived from core OCE326-GGC5[6], time series from 20 to 27 ka BP derived from ODP Site 1063[26]. Yellow bars = phases of increases in subSST$_{Mg/Ca}$ into Heinrich Events and the Younger Dryas (YD); grey bars = Heinrich Layers (HE2 Heinrich Event 2, HE1 Heinrich Event 1) and the YD. B/A Bølling–Allerød interstadial, HS Heinrich Stadials.

difference may stem from the fact that the Younger Dryas is not a typical Heinrich Event with only modest and non-linear fresh-water forcing (and modest AMOC weakening), proposed to be delivered to the ocean via, e.g., the St. Lawrence River[27] or Mackenzie River[28].

The rapid North Atlantic subsurface ocean warming during AMOC weakening is a feedback mechanism observed in several numerical modelling simulations[29–31]. As soon as the AMOC is weakened, due to freshwater hosing, immediate cooling and isolation of the surface ocean from subsurface waters is observed in the models[29–31]. Physically, reduced convection of North Atlantic Deep Water leads to a redistribution of heat in the Atlantic basin and the interior-ocean warms, in particular in the equatorial- and South Atlantic[29]. Notably, some models exhibit ocean warming in the Atlantic basin down to ~2500 m water depth within decades after AMOC slowdown[30,31]. At the same

time, subsurface advection of warm waters by the subtropical and subpolar gyres are important for the poleward heat transport to the North Atlantic. As long as deep-water convection weakens in the North Atlantic, the surface ocean cools in the subpolar Atlantic. The subsurface ocean further warms because of the background energy transport of the gyre system together with strong isolation from freshening of the surface ocean, thereby creating a growing temperature inversion[31]. The magnitude of interior-ocean warming strongly depends on the used model, as well as on model configurations and varies between 4 and 6 °C[30,31], consistent with our subSST$_{Mg/Ca}$ reconstructions. More-over, as soon as the AMOC recovers from freshwater forcing with active deep convection in the North Atlantic, the surface ocean becomes warmer and interior-ocean temperatures become cooler due to invigorated AMOC[31]. Our records verify and expand results from theoretical modelling experiments[29–31], and we

propose subsurface ocean warming of the western subpolar North Atlantic is a natural feedback mechanism to slowdowns of the AMOC over the last 27,000 years.

**Subsurface ocean warming as trigger for Heinrich Events.** Several ice-sheet modelling studies propose North Atlantic subsurface warming as a trigger for increased ice-rafting[9,11–13,32]. Accordingly, the build-up of ocean heat near the grounding line of ice-shelves is a critical value in some of these models to trigger a rapid retreat of the ice-margin around the Labrador Sea[9,13,32]. With continuous subsurface ocean warming the ice-shelf shrinks and accelerates the ice flow at the grounded line, triggering the rapid surge of the grounded ice and a massive iceberg discharge in the modelling simulations[13]. A more recent ice-sheet modelling study showed that without an ice-shelf relatively small fluctuations in $subSST_{Mg/Ca}$ near Hudson Strait are sufficient to trigger Heinrich Events[11]. Our findings are in close agreement with ice-sheet modelling studies, providing first solid evidence for subsurface ocean warming in the western subpolar North Atlantic as the trigger for ice-sheet instabilities during Heinrich Events[9,11–13,32].

**Future AMOC slowdown and the build-up of ocean heat in the subpolar Atlantic.** Instrumental time-series show a long-term increase in the North Atlantic heat content since the early 1950s and a close relationship to the accelerated mass loss of the Greenland Ice Sheet[33–35]. However, it is yet unclear how AMOC variability contributes to the observed changes because of analytical limitations from relatively short instrumental time-series. The most recent report of the Intergovernmental Panel on Climate Change projects a future decline of the AMOC due to anthropogenic warming in the 21st century[36]. New empirical data suggest the AMOC has been evolved to a point close to a critical transition to its weak circulation mode[37]. Our findings suggest that past critical transitions to a weak AMOC mode are accompanied by the massive build-up of ocean heat in the western subpolar North Atlantic triggering ice-sheet instabilities during Heinrich Events. The projected weakening of the AMOC in the 21st century[36] may result in an amplified increase in the interior-ocean heat content that could be critical for the stability of modern, marine-terminating Arctic glaciers and the freshwater budget of the North Atlantic.

## Methods

**Mg/Ca and δ18O measurements.** For Mg/Ca analyses, ~100 individuals of the foraminiferal species *N. pachyderma* sin. were picked from sieved sediment samples of the size fraction >250 μm. Prior to geochemical analyses, the foraminiferal tests were opened gently between glass plates and samples split into one third for stable isotope analyses and two third for Mg/Ca measurements. Additional cleaning of each sample was done according to ref. [38]. Mg/Ca analyses were performed with an axial-viewing ICP-OES Varian 720 ES (Inductively Coupled Plasma Optical Emission Spectrometry) at GEOMAR, Helmholtz-Centre for Ocean Research in Kiel. The levels of detection typically vary for each cation, ranging from 0.001 to 0.1 μg/ml. Mg/Ca was normalized to the ECRM 752–1 standard (3.761 mmol/mol Mg/Ca; according to ref. [39]) and drift-corrected. Regular analyses of the ECRM 752–1 standard yield an analytical error of ±0.01 mmol/mol for Mg/Ca.

Simultaneous measurements of Fe and Al were performed to monitor possible silicate contamination[38]. Six out of 110 samples showed elevated Al/Ca values (>0.2 mmol/mol), however, are inconspicuous in the Mg/Ca value compared to neighbouring Mg/Ca values with no sign of potential contamination (Supplementary Fig. 1a). We identified one sample with an extremely high Al/Ca value of ~1.2 mmol/mol. A replicate measurement of this sample showed a very low Al/Ca value of ~0.04 mmol/mol, no sign of contamination, and nearly identical Mg/Ca values (1.93 mmol/mol with elevated Al/Ca value; 2.01 with low Al/Ca value). Obviously, the elevated Al/Ca values do not largely afflict the Mg/Ca values in the sample. A cross-plot of Mg/Ca values versus Al/Ca values further confirms the generally weak correlation between Mg/Ca and Al/Ca ($r^2 = \sim 0.14$; Supplementary Fig. 1b). Consequently, we decided to report all values. We note that discarding the six values with elevated Al/Ca values would not change the observed trends in the discussed time-series (Supplementary Fig. 1a).

Stable isotopes (δ18O) of *N. pachyderma* sin. were measured with Thermo Scientific MAT 253 mass spectrometers equipped with an automated Kiel IV Carbonate Preparation Device at MARUM, University of Bremen and at GEOMAR, Helmholtz-Centre for Ocean Research in Kiel. The δ18O isotope values were calibrated versus the NBS19 (National Bureau of standards) carbonate standard and an in-house standard ("Standard Bremen"). The long-term analytical precision is 0.06‰ for δ18O. Results were calibrated to the VPDB scale, given in per mille (‰) relative to the VPDB.

**Mg/Ca temperatures of *N. pachyderma* sin.** The foraminiferal Mg/Ca ratios were converted into water temperatures considering different species-specific calibrations for *N. pachyderma* sin.:

$$Mg/Ca(mmol/mol) = 0.474 \exp(0.107 * T) \tag{1}$$

$$Mg/Ca(mmol/mol) = 0.13(\pm0.037) * T(degC) + 0.35(\pm0.17) \tag{2}$$

$$Mg/Ca(mmol/mol) = 0.4 * \exp(0.1 * T) \tag{3}$$

Comparison of converted core-top Mg/Ca temperature to instrumental temperatures[20] allowed assessment of the most suitable calibration for our study site (Supplementary Fig. 2). Assuming a mean habitat depth of 150 m for subsurface dwelling species of *N. pachyderma* sin[40] yields a difference of 1.2 °C after calibration to Eq. 1 (ref. [41]), 0.18 °C after calibration to Eq. 2 (ref. [42]) and 4.32 °C after calibration to Eq. 3 (ref. [43]). Consequently, we considered the calibration after ref. [42] as appropriate to convert the Mg/Ca ratio to subsurface temperatures.

**Sensitivity of *N. pachyderma* sin. derived Mg/Ca signal to changes in habitat depth.** A range of habitat depths from ~50 to 200 m has been reported for *N. pachyderma* sin.[40,42]. Consequently, we checked whether changes in the assumed habitat depth may afflict the robustness of our results. To do so, we calculated the temperature gradient between 50 and 200 m water depth ($\Delta T_{200-50}$) using modern instrumental data[20] from a water column profile near-site GeoB18530-1 (Supplementary Fig. 3). The $\Delta T_{200-50}$ is 2.05 °C, assuming a maximum shift in the habitat depth of *N. pachyderma* sin. in September. Given the large subsurface temperature amplitudes of up to 6 °C observed in our time series we conclude that only a small part of the total variability is explained by changes in the habitat depth.

**Sensitivity of *N. pachyderma* sin. derived Mg/Ca to changes in the seasonal cycle.** We further investigated the effect of major shifts in the seasonal cycle of *N. pachyderma* sin. on reconstructed subsurface temperatures. A study combining sediment trap data and modelling results suggests that the seasonality in the production of *N. pachyderma* sin. may have changed by up to 6 months between the Last Glacial Maximum (LGM) and full Heinrich conditions in the western subpolar North Atlantic[44]. However, we note that reconstructed subsurface warmings preceding Heinrich Events in our time series and thus, do not reflect full Heinrich conditions. Nevertheless, we investigated the effect of a possible change in seasonality assuming a maximum shift in seasonality from April to September, considering modern hydrographic data[20]. The difference in $subSST_{150m}$ between April (7.2 °C) and September (7.45 °C) is 0.25 °C at our core site (see Supplementary Fig. 3). We conclude that the effect of a seasonal shift in production of *N. pachyderma* sin. is minor on reconstructed subsurface temperatures.

**Oxygen isotopic composition of seawater (δ18O$_{ivc-sw}$).** We calculated the regional ice-volume-corrected δ18O$_{sw}$ record (δ18O$_{ivc-sw}$) considering changes in global δ18O$_{sw}$ due to continental ice-volume variability using the relative sea-level curve of ref. [45]. To remove the temperature effect from the δ18O$_{ivc-sw}$ record we applied the temperature versus δ18O$_{calcite}$ equation of ref. [46]:

$$\delta^{18}O_{calcite}(‰V - PDB) = (21.9 - 3.16 * (31.061 + T(°C))0.5) + \delta^{18}O_{sw}(‰V - PDB) \tag{4}$$

The calculated δ18O$_{ivc-sw}$ were converted from V-PDB into Vienna Standard Mean Ocean Water (V-SMOW) scale according to equation of ref. [47]:

$$\delta^{18}O(‰V - PDB) = 0.9998 * \delta^{18}O_{sw}(‰V - SMOW) - 0.27‰ \tag{5}$$

Low (high) values of δ18O$_{ivc-sw}$ pointing to fresh (saline) ocean conditions.

We performed an error propagation analysis to assess the error of the δ18O$_{ivc-sw}$ calculations considering the uncertainty in performed Mg/Ca and δ18O measurements, the uncertainty of applied Mg/Ca calibration as well as the error involved in the relationship between salinity and δ18O$_{sw}$. We obtained an error of ~0.38 for reconstructions of δ18O$_{ivc-sw}$. A previous study reported similar error propagations for δ18O$_{sw}$ of 0.36 (+/−0.02%) derived from δ18O$_{Calcite}$ and Mg/Ca-temperature reconstructions of *N. pachyderma* sin. in the subpolar Pacific[48].

**X-ray fluorescence scanning.** X-ray fluorescence core-scanning measurements of GeoB18530-1 were performed with an Avaatech XRF Core Scanner at the MARUM, University of Bremen. Elemental intensities were obtained at 1 cm resolution over a 1.2 cm² area from the split core surface of the archive halves. The

split-core surface was scanned three times with different settings for light (e.g., Al, Ca; 10 kV, 20 s, 150 mA), medium (e.g. Sr, 30 kV, 20 s, 150 mA) and heavy (e.g. Ba; 50 kV, 20 s, 800 mA) elements. To avoid contamination of the XRF measurement unit the core surface was covered with a 4 µm thin *SPEXCerti Prep Ultralene* foil. The core scanner unit includes a *Canberra X-PIPS* Silicon Drift Detector (SDD; Model SXD 15C-150-500) with 150 eV X-ray resolution, a *Canberra* Digital Spectrum Analyser DAS 1000 and an *Oxford Instruments* 100 W Neptune X-ray tube with rhodium (Rh) target material. Raw data spectra were processed by Analysis of X-ray spectra by the Iterative Least square software (WIN AXIL) package from *Canberra Eurisys*.

**Age models**. The age model for gravity core GeoB18530-1 is based on 20 accelerator mass spectrometry radiocarbon dates from samples of *N. pachyderma* sin. (19 samples) and one mixed sample (*Globigerina bulloides* and *N. pachyderma* sin.) from core GeoB18530-1 (Supplementary Table 1). Radiocarbon ages were measured at Poznań Radiocarbon Laboratory as well as Physics Institute, University of Bern Climate and Environmental Physics Radiocarbon lab. The age model is based on an ensemble of 2.000 age-depth realizations calculated with the age modelling software BACON[49]. Radiocarbon ages were calibrated using the IntCal20 calibration curve[50] and modelled local reservoir ages[51] provided through the toolbox PaleoDataView[52]. To assess the uncertainty of Mg/Ca temperatures, we combined 2000 noisy proxy realisations with the 2000 BACON age models, interpolated the resulting 2000 time series to the median ages of all sampling depths and calculated the 95% confidence envelope for the Mg/Ca SSTs. We further re-calibrated the reported radiocarbon ages from site MD01-2461[24] following the same procedure as described for core GeoB18530-1.

## Data availability

All relevant data in this paper are available at PANGAEA Data Publisher (https://doi.org/10.1594/PANGAEA.943563).

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

## Acknowledgements
Sample material has been provided by the GeoB Core Repository at the MARUM - Center for Marine Environmental Sciences, University of Bremen. This research used data acquired at the Core Scanner Lab at MARUM - Center for Marine Environmental Sciences, University of Bremen. We thank V. Lukies for the technical support in the Core Scanner Lab. N. Gehre is acknowledged for lab support and preparation of Mg/Ca and stable isotope measurements. L.M. wishes to extend his special thanks to C. Kricke, H. Max and L. Max for their continuous and unparalleled patience and support. L.M. received financial support from the Deutsche Forschungsgemeinschaft (DFG) through grant 47010623 (SAINT). C.M.C. acknowledges the financial support from FAPESP (grants 2018/15123-4 and 2019/24349-9), CAPES (grant 88881.313535/2019-01), CNPq (grant 312458/2020-7) and the Alexander von Humboldt Foundation.

## Author contributions
L.M. and S.M. designed the study, L.M collected the XRF data, L.M. and S.M. provided radiocarbon data, L.M. and M.M.L. prepared and analysed δ18O data, D.N. facilitated Mg/Ca measurements and additional stable isotope measurements. C.M.C. contributed to sample collection and background knowledge. L.M., D.N., C.M.C. and S.M. wrote the first draft of the manuscript. All authors contributed to the interpretation and the preparation of the final manuscript.

## Funding

## Competing interests
The authors declare no competing interests.
