## [Peer Review File · Nature Communications]

Subsurface ocean warming preceded Heinrich EventsReviewers' Comments:

Reviewer #1:

Remarks to the Author:

Review of Subsurface-ocean warming preceded abrupt ice-sheet instabilities over the past 30,000 years by Max et al

The submission offers some interesting evidence in support of sub-surface ocean warming occurring just before IRD increases in one new marine record over the last @ 27 kyr, suggesting a causal role. However, as detailed below, from the title to the end, the submission is unfortunately characterized by overstatements, misleading statements, incorrect statements, and imprecise language. Potentially significant uncertainties are inadequately addressed if not ignored. This could be a worthwhile submission for Nature Communications, but not in its current form.

title

Subsurface-ocean warming preceded abrupt ice-sheet instabilities
2 over the past 30,000 years

This is misleading since the record only goes back to @ 27 ka, with the oldest AMS C14 date at only @ 26.5 ka. The core lacks HE 3 which occurred about 30 ka

23 of ocean heat near the critical depth for basal melting of the Laurentide Ice Sheet

The majority of the LIS (Laurentide Ice Sheet) was terrestrial, so above statement makes no sense due to imprecise wording

highlighting the

27 paramount role of ocean circulation changes on driving past ice-sheet
28 instabilities.

This is stated as a fact, but I should not have to remind that correlation is not causation...

28 Numerical modeling simulations predict a future decline of the

all, some, or just one simulation?

Our findings imply that the projected slowdown in Atlantic

31 Meridional Overturning Circulation will play a central role for the ocean heat
32 budget in the subpolar North Atlantic

That's a hell of a claim based on one new marine core.

global ice

44 volume-free oxygen isotopic composition of seawater

ice volume-free is confusing, how about "ice-volume removed"

56 against major instabilities of the Laurentide Ice Sheet (LIS) because of

Do you mean instabilities of Hudson Bay ice stream or what exactly? The LIS was a very large.

Moreover, sediment core GeoB18530-1

55 is ideally located to study the relative timing of past North Atlantic climate variability
56 against major instabilities of the Laurentide Ice Sheet (LIS) because of: (i) its location
57 close to the exit of the Labrador Sea, the North Atlantic gateway of main iceberg
58 trajectory associated with Hudson Strait Heinrich Events;

I find this statement arguable especially since fig 1 is a bit misleading as it shows present-day circulation of the Labrador current around Newfoundland, right through the Grand Banks that would have been terrestrial during much of MIS2 (for sure during HE2). From a circulation point of view, it's not clear to me how well the core site will be exposed to icebergs originating from Hudson Strait. What about StL?

fig 2 doesn't go back to 30 ka, as H3 is not present. The paper is also lacking any provenance analysis, so unclear if IRD is from Hudson Strait or Gulf of St. Lawrence.

I'm also confused why many of the datapoints in the upper curves of fig 2 have no error bars.

Is high Ca/Sr a definitive indicator of Hudson Bay/Strait Paleozoic limestone and dolostone? A marine geologist colleague indicated to me that Baffin Bay sources also have this characteristic. What would IRD from Gulf of St. Lawrence give on Ca/Sr? More detailed provenance assessment would significantly improve the submission.

In this regard, we observe

77 subSST maxima synchronous with the onset of IRD deposition during Heinrich
78 Events at site GeoB18530-1 (Fig. 2b).

The maxima inferred warming at the start of HE2 is pretty weak and the strongest maximum in the record is at the end of HE1, counter to the causal claim of the paper. This issue might be addressable, but has not been within the text.

fig 3: The yellow ocean warming colouring is problematic as the strongest indicated warming occurs that the end of HE1 with blue colouring.

which are related to

94 two different ice-stream advances¹⁸.

Fact or inference?

Overall, these findings unequivocally exhibit the

95 close relationship between subsurface ocean temperature increases and the beginning
96 of Heinrich Events in the North Atlantic over the last 30,000 years (Fig. 3).

"unequivocally" is an overstatement given the issues I've raised above

further crossing a
111 critical threshold and destabilized the British Ice Sheet²¹.

Fact or inference?

The quasi-simultaneous rise of subSST in both
115 margins of the subpolar North Atlantic implying a basin-wide built-up of ocean heat
116 that began at ~18.5 ka BP (Fig. 4b).

results from 2 cores "imply" basin-wide build-up?????

This is largely consistent with a basin-wide
117 increase of North Atlantic bottom water temperatures observed prior to Heinrich
118 Event 1²²

240²² Marcott, S. A. et al. Ice-shelf collapse from subsurface warming as a trigger for Heinrich
events.
241 Proc. Natl Acad. Sci. 108, 13415–13419 (2011).

"observed" or inferred?

We therefore propose that an enormous volume of heat was stored in the
119 ocean-interior of the subpolar North Atlantic at this time.

another causal issue that is avoided. Even if the above claim is true, how
do you get the heat up into the Northern part of the Labrador sea to reach
Hudson Strait if this were the instability region (or is the instability region
Gulf of St. Lawrence or ??? The boundary current for the Labrador Sea is largely
fed by the Eastern Greenland current, at least presently.

121 Third, proxy records of AMOC variability^{9,23} show that the deep ocean circulation
122 weakened at the beginning of Heinrich Stadial 2 and 1, preceding Heinrich Events by
123 several hundreds of years (Fig. 4d).

without showing age uncertainties, how is one supposed to evaluate this statement?

128 fact that the Younger Dryas is not characterized by a typical Heinrich Event with only
129 and non-linear freshwater forcing delivered to the ocean via either the St.
130 Lawrence River²⁴ and the Mississippi River²⁵ or the Mackenzie River²⁶. Nonetheless,

The 1975 vintage reference²⁵ would not have had AMS C14 dating. More recent records
show clear shutdown of Mississippi discharge before the start of the Younger Dryas.

137 The rapid North Atlantic interior-ocean warming during AMOC weakening is a
138 feedback mechanism also observed in several modeling simulations²⁷⁻²⁹. As soon as
139 the AMOC is suppressed, due to freshwater hosing, immediate cooling and isolation
140 of the surface ocean from subsurface waters is observed in the models²⁹

Why is only²⁹ cited in the second sentence if the claim in the first sentence is
for all 3²⁷⁻²⁹ studies?

Physically,

141 without cooling by convection, the ocean-interior warms, in particular in the
142 equatorial- and South Atlantic³⁰.

Should make clear what heat source you are talking about. Ocean interiors don't just warm on their own.

Moreover, as soon as the AMOC

151 recovers from freshwater forcing, the surface ocean becomes warmer and subsurface
152 ocean temperatures cooler due to active convection

convection where? Deep convection occurs in the high latitudes, still really confused. Do mean AMOC? THC? or?

155 Our proxy time-series now provide solid evidence for the built-up of an ocean heat
156 reservoir in the North Atlantic that occurred during AMOC slowdowns over the last
157 30,000 years

Over what exact geographic region? Would help interpret the claims presented.

159 Several modeling studies discussed the substantial role of

What kind of "modelling studies"?

Another major point is the water depth for the terminus of

166 ice-shelves to be approximately 300 m for the LIS, analogue to modern Greenland,
167 Svalbard and Alaskan marine-terminating-glaciers³³ We now provide the first
168 indication yet for the proposed ocean-interior warming near the critical depth to
169 initiate basal melting and further destabilization of the LIS, culminating into Heinrich
170 Events.

Water temperature at depth of terminus of ice shelf is not going to have much relevance for melt under an ice shelf. Water temperature at the deeper grounding line relative to upper level temperature is much more critical for driving sub-shelf melt, cf bouyant plume modeling papers of sub-shelf melt.

Our study reveal a fundamental relationship between AMOC strength and
185 ocean heat content in the North Atlantic

overstatement

Our findings imply that a near future decline

186 in AMOC will cause rapid increases of the North Atlantic heat content with
187 unparalleled magnitude compared to instrumental time-series

overstatement

404 Seawater approximating paleo-salinity ($\delta^{18}\text{O}_{\text{sw-ivf}}$)

405 We calculated the regional ice volume free $\delta^{18}\text{O}_{\text{sw}}$ record ($\delta^{18}\text{O}_{\text{ivf-sw}}$) consider..

so relying on single inversions with significant uncertainties that are ignored

Reviewer #2:

Remarks to the Author:

Review of "Subsurface-ocean warming preceded abrupt ice-sheet instabilities over the past 30,000 years", by Max et al.

This manuscript describes collected proxies of subsurface temperature and IRD of the last 30 kyrs in a sediment core located in the southwest subpolar North Atlantic Ocean. These proxies show a consistent pattern of subsurface warming prior to the IRD arrival during Heinrich events and the Younger Dryas. Because this is done on the same core, there can not be doubts about the relative timing of the shown variables.

The analysis of the time series argues in favor of the interpretation of Heinrich events being triggered through an induced destabilization of the Laurentide ice sheet by warming of the ocean in contact with its basal layer.

The current piece of work, together with Barker et al. 2015 nicely contribute, in my opinion, to the ultimate demonstration that Heinrich events were triggered through oceanic changes. And not the opposite, as it has been thought by a large part of the community for many years.

The manuscript is very well written, it reads fluently and the figures clearly support the description and interpretation of the presented proxies.

I recommend publication, essentially "as it is".

.....

I only have one minor suggestion regarding the discussed implications of their work for the current climate (at the end of the abstract and conclusions):

It is true that the AMOC is likely to be already suffering a strong decrease in its intensity (as shown by several papers), and that, as you show here, this has implications for the inner ocean heat reservoir. That being said, it is also known that the Greenland ice sheet is relatively "disconnected" to the ocean. The main factor for its decreasing mass in the coming centuries will be an increasingly negative surface mass balance. Although the warming waters of the fiords having a glacier terminy will indeed facilitate the grounding line migration and therefore enhance ice discharge to the ocean, this process will not be the dominant one for Greenland.

Therefore, I propose that the authors lower the supposed importance of their findings for understanding the future of the Greenland ice sheet announced in the abstract and at the end of the manuscript.

This can be done directly by rephrasing those parts of the manuscript, or it could also be done by expanding that line of thoughts regarding the future climate to the implications for Antarctica.

It is clear that the stability of the Antarctic ice sheet is directly linked to the Southern Ocean behavior during next centuries (see for example Golledge et al. 2015, Nature). What it is not so clear is the implications of a decreasing or halted AMOC on the Southern Ocean subsurface temperatures and thus the final implications on Antarctica.

If the authors want to keep their interpretation of the implications that their work has on future sea level rise, I recommend them to expand on the aforementioned issue.

Reviewer #3:

Remarks to the Author:

The authors present high resolution proxy data for a core from the subpolar N Atlantic. The methods employed are generally "traditional". A big advantage of the current study is the high-resolution records from a very nice core. The data allow them to resolve timing of various tracers associated with Heinrich events over the past ~30 ka. Possible links between subsurface warming and icesheet instabilities are discussed.

To get the data published in Nature Communications, more work is needed. I have listed my comments in details below, and I hope they are useful for the authors to revise their manuscript.

Writing requires substantial improving.

1. age model and timing of events

Timing is very critical for the conclusion of this paper. Although the relative timing of Mg/Ca and Ca/Si is well defined (Fig. 2), it is necessary to know when changes in Mg/Ca & Ca/Se occurred in the context of N Atlantic climatic events (Fig. 4a). That is, a robust calendar age model is needed.

Currently, the age model is constructed using model-based surface reservoir ages. I don't think this is a good/robust practice. The problematic age model is reflected by asynchronous warming at SU8118 (and possibly MD01-2461) and NGRIP at Bolling onset and YD termination and possibly at HS2 termination (Fig. 4). For example, how can warming at SU8118 lag by so long time relative to NGRIP d18O at Bolling onset and YD termination? Unless the authors have a sound interpretation for such warming offsets in timing, the age model needs to be reconsidered.

I suggest trying Nps% to obtain an independent age model for the core. Nps% could be useful to pin down timing for YD, HS2, and Bolling onset (see Thornalley et al., 2011).

It is necessary to know when warming and Ca/Sr changes occurred, including starting and ending dates. For example, was warming started during a certain stadial or at the end of the preceding interstadial?

A robust age model can be valuable to reveal possible new mechanisms controlling stadial icesheet instabilities.

2. Icesheet instability indicators

The use of Ca/Sr as a tracer to reflect icesheet instability is implicit, but needs to be explicitly spelled out in the main text.

Also, does it mean that icesheet is stable if Ca/Sr values remain low?

Additionally, some secondary Ca/Sr peaks are seen (Fig. 2c; 430 cm, 150 cm, etc); do they also reflect icesheet instabilities? What is the threshold to be used to reflect an icesheet instability? Please define a value in Ca/Sr. This seems arbitrary as it stands.

3. Mechanisms for the early warming and icesheet stability

What's new that can be learnt from the data presented? Is there any new mechanism for the icesheet instability, and for the warming prior to the icesheet instabilities?

My impression is that the new data provide evidence to support subsurface warming that may cause icesheet melting during Heinrich stadials. The related ideas have long been published (e.g., ref 22). To get this work published in Nature Communications, I encourage the authors to think deeply about new

findings that can advance our understanding of icesheet stability and abrupt climate changes in the North Atlantic (and broader) regions.

Some large increases in Mg/Ca are observed at other times such as 420 cm, 200 cm, etc (Fig. 2); what caused these warming events? Also, these warming events do not follow large increases in Ca/Sr (that is, icesheet instability). For HS2, we see double warming prior to the large Ca/Sr peak. In contrast to the warming around 25.5 ka which was followed by large Ca/Sr, warming at ~26.5 ka (of similar magnitude as occurred at 25.5 ka) did not "initiate" any Ca/Sr rise. Do these observations suggest that warming does not always initiate instability of icesheets? If so, implications for future climate would be ambiguous, and revision is necessary (e.g., Abstract and the last paragraph).

4. Data processing and proxy interpretation

Statistical analyses – Monte Carlo simulation of reconstructions are necessary for records shown in Fig. 4, especially for records generated in this study.

Additionally, what is the point of showing $\delta^{18}O_{ivf-sw}$? The data appear to be not well used.

Detailed comments:

Abstract: unless the authors can provide model results for future scenarios, implications for the future should be brief and limited to one sentence.

Main: It starts very abruptly into data/methods. I strongly suggest adding one opening paragraph to describe the current status of related studies and why this study is warranted.

Line 60-62: correct grammar.

Line 64-65: how to define the timing mentioned here? When/where is the onset of HEs? See my concerns above about using model-based surface reservoir ages to convert $14C$ to calendar ages. Is YD treated as a HE? If so, where is the onset of YD? can the small Ca/Sr peak at ~335 cm be treated as the start of IRD rise? If so, why cannot other similarly small Ca/Sr rises (e.g., 220 cm, 440 cm, etc) be treated as IRD rises (and icesheet instabilities)? A value of Ca/Sr needs to be defined to detect IRD events.

Line 87-90: based on Fig. 4c, we do see some warming coeval with subSST rise (Fig. 4b) around HE1 (17.5 ka) and YD, inconsistent with descriptions here.

Paragraph starting from line 95: description unclear; substantial improvement is necessary.

Line ~116: if one looks into Figs. 3 and 4 (which is difficult), the Ca/Sr peak labelled as "YD" in Fig. 3 appears to correspond to a time after YD shown in Fig. 4 (corresponds to PB?). I suggest adding Ca/Sr in Fig. 4 to facilitate comparing of relative timing of various records. Currently, it is very difficult, if not possible, to read two figures back and forth.

Line 121: "a fundamental relationship" – what is it? Specify.

Line 140-142: reference? Note that CESM has long simulation runs.

Line 82-84: "close relationship", please specific what the relationship is.

Line 157: "paramount role", "fundamental conclusion", so far, similar wording appears several times. Please demonstrate the role and what is new in terms mechanisms that gleaned from the new records! Otherwise, avoid exaggerating the claim.

Line 171: again "fundamental relationship"... explain- what is it?

Line 328-335: Al/Ca detection limit? Add unit (mmol/mol) after Al/Ca and Mg/Ca values wherever necessary.

Fig. 4: add Ca/Sr in the figure to facilitate comparisons; add yellow bars to indicate warming prior to ice-sheet instabilities (Ca/Sr). Add age control points in the figure as well. For HS2, AMOC recover led NGRIP warming – how cause this large leading, or is it an age model issue?

Thornalley, D. J. R., Barker, S., Broecker, W., Elderfield, H. & McCave, I. N. The Deglacial Evolution of North Atlantic Deep Convection. *Science* 331, 202-205 (2011).

**Author response to “Subsurface-ocean warming preceded abrupt ice-sheet instabilities over the past 30,000 years” by L. Max et al.
NCOMMS-21-35783-T**

Response to comments by Reviewer #1:

We thank Reviewer #1 for the comprehensive and constructive comments.

Summary:

The submission offers some interesting evidence in support of sub-surface ocean warming occurring just before IRD increases in one new marine record over the last @ 27 kyr, suggesting a causal role. However, as detailed below, from the title to the end, the submission is unfortunately characterized by overstatements, misleading statements, incorrect statements, and imprecise language. Potentially significant uncertainties are inadequately addressed if not ignored. This could be a worthwhile submission for Nature Communications, but not in its current form.

Comments and suggestions:

*2 Subsurface-ocean warming preceded abrupt ice-sheet instabilities over the past 30,000 years.
This is misleading since the record only goes back to @ 27 ka, with the oldest AMS C14 date at only @ 26.5 ka. The core lacks HE 3 which occurred about 30 ka.*

We agree. It is true that Mg/Ca subsurface temperatures only go back to ~27 ka BP. To avoid confusion we changed the title to “Subsurface ocean warming preceded abrupt ice-sheet instabilities during Heinrich Events”.

*23 of ocean heat near the critical depth for basal melting of the Laurentide Ice Sheet
The majority of the LIS (Laurentide Ice Sheet) was terrestrial, so above statement makes no sense due to imprecise wording*

We agree. We revised our statement, now pointing out we are talking about melting of marine-terminating portions of the LIS (lines 20 – 23).

27-28 highlighting the paramount role of ocean circulation changes on driving past ice-sheet instabilities.

This is stated as a fact, but I should not have to remind that correlation is not causation...

We toned down our statement about links between ocean circulation and ice-sheet instabilities (lines 23 - 26).

*28 Numerical modeling simulations predict a future decline of the
all, some, or just one simulation?*

In the original manuscript we referred to ensemble model results (27 models considered) presented in the IPCC, 2019: IPCC special report on the ocean and cryosphere in a changing climate, dealing with the future behavior of the AMOC.

However, we rephrased the implications of our study following the suggestions of Reviewer #2 (lines 26 – 30).

31–32 *Our findings imply that the projected slowdown in Atlantic Meridional Overturning Circulation will play a central role for the ocean heat budget in the subpolar North Atlantic*
*That's a hell of a claim based on one new marine core.*

We consider our results are very robust and support our conclusions. However, having one marine record only we decided to tone down our statement to avoid exaggerating the claim (lines 26 – 30).

44 *global ice volume-free oxygen isotopic composition of seawater*
*ice volume-free is confusing, how about "ice-volume removed"*

Both ice-volume-free as well as ice-volume-corrected have been used in similar studies but not “ice-volume-removed”. We decided to stay with the terminology given by e.g. Nürnberg et al. (2021).

56 *against major instabilities of the Laurentide Ice Sheet (LIS) because of*
*Do you mean instabilities of Hudson Bay ice stream or what exactly? The LIS was a very large.*

As we stated in lines 57-58 in the original manuscript we refer to “Hudson Strait Heinrich Events” We revised the paragraph making this point more clear (lines 78 - 81).

55-58 *Moreover, sediment core GeoB18530-1 is ideally located to study the relative timing of past North Atlantic climate variability against major instabilities of the Laurentide Ice Sheet (LIS) because of: (i) its location close to the exit of the Labrador Sea, the North Atlantic gateway of main iceberg trajectory associated with Hudson Strait Heinrich Events;*

*I find this statement arguable especially since fig 1 is a bit misleading as it shows present-day circulation of the Labrador current around Newfoundland, right through the Grand Banks that would have been terrestrial during much of MIS2 (for sure during HE2). From a circulation point of view, it's not clear to me how well the core site will be exposed to icebergs originating from Hudson Strait. What about StL?*

We agree that large parts of the Grand Banks have probably been terrestrial under glacial conditions. However, we would expect the position of the glacial Labrador Current far more offshore (and closer to our core site) because of larger terrestrial exposure of the Grand Banks.

1. In general, Heinrich Layers are a typical feature of glacial sediments in the North Atlantic and have been traced from roughly 40°N to 55°N, even reaching into the central and eastern parts of the North Atlantic (e.g. Hemming, 2004). Site GeoB18530-1 is located at the southwestern tip of the well-documented IRD-belt that depicts the main trajectory of icebergs associated with Heinrich Events in the North Atlantic (revised Fig. 1b).

2. The use of Ca/Sr as a tracer for Heinrich Events is well-established and has been extensively used to identify and correlate Heinrich Layers throughout the North Atlantic (e.g., Channel et al., 2012). The geochemical signature of IRD with elevated Ca/Sr ratios is a typical feature for Paleozoic limestone and Dolostone originating from the Hudson Strait area (Hodell et al., 2008; 2017). We revised the paragraph pointing out more clearly the use of Ca/Sr as an indicator for Hudson Strait Heinrich Events (lines 92 - 95).

3. The onset of IRD deposition (HE1 and HE2) preserved in core GeoB18530-1 is well constrained via AMS ¹⁴C ages in close agreement with known timing of Hudson Strait Heinrich Events elsewhere in the North Atlantic (e.g. Bond et al., 1992 or Hodell et al., 2017), discussed in lines 103 – 106; 115 – 120.

4. We further compared core GeoB18530-1 to records from cores GeoB18514-2 and GeoB18515-2 from the Laurentian Fan, under the influence of the St. Lawrence River (figure below; Mulitza et al., 2015).

The sediment cores from the Laurentian Fan feature several “reddish mud beds” from 22 to 17 ka BP that are many meters thick and very different to IRD deposition at core site GeoB18530-1. The deposition of the reddish mud beds is interpreted as “deglacial outburst floods” found across a large area of the Laurentian Fan and the adjacent continental slope (Leng et al., 2018).

fig 2 doesn't go back to 30 ka, as H3 is not present. The paper is also lacking any provenance analysis, so unclear if IRD is from Hudson Strait or Gulf of St. Lawrence.

According to Leng et al. (2018) Gulf of St. Lawrence sediments feature magnetic properties which represent two magnetite-rich sources (i.e., the Canadian Shield and/or the southern Newfoundland) and two hematite-rich sources (i.e., Appalachian red beds). Thus, we exclude a dominant influence of the Gulf of St. Lawrence on the sedimentation at site GeoB18530-1 between 22 and 17 ka BP.

I'm also confused why many of the datapoints in the upper curves of fig 2 have no error bars.

We added the error of Mg/Ca SSTs at the top left of Fig. 2b.

Is high Ca/Sr a definitive indicator of Hudson Bay/Strait Paleozoic limestone and dolostone? A marine geologist colleague indicated to me that Baffin Bay sources also have this characteristic. What would IRD from Gulf of St. Lawrence give on Ca/Sr? More detailed provenance assessment would significantly improve the submission.

We agree. Sediments from Baffin Bay area also contain Paleozoic Limestone and Dolostone and we cannot exclude IRD signatures from Baffin Bay, which is very proximal to Hudson Strait. Nevertheless, in case of Baffin Bay we would expect similar transport and IRD deposition via icebergs to site GeoB18530-1 and this will not affect our conclusions. However, we exclude that IRD in GeoB18530-1 are dominated by material from the Gulf of St. Lawrence (please see our detailed comments about Gulf of St. Lawrence sediment characteristics above).

77-78 In this regard, we observe subSST maxima synchronous with the onset of IRD deposition during Heinrich Events at site GeoB18530-1 (Fig. 2b).# The maxima inferred warming at the start

of HE2 is pretty weak and the strongest maximum in the record is at the end of HE1, counter to the causal claim of the paper. This issue might be addressable, but has not been within the text.

The subSST increased by ~5°C during the transition to HE2 (Fig. 3). Thus we do not agree that the warming signal is weak. However, we agree that the HE2 warming is weaker compared to the warming during transition to HE1 (Fig. 3).

Hodell et al. (2017) looked a little bit more into the details of IRD deposition during HE1 using a core from the central North Atlantic (IODP U1308). They found that HE1 is not a single IRD event but rather consists of two distinct pulses of IRD deposition (Heinrich Event 1.1 and 1.2), further suggesting two stages of ice-stream advances from Hudson Strait or two different ice-streams. The double peak in subSST at the beginning and end of HE1 in core GeoB18530-1 might be related to Heinrich Event 1.1 and 1.2 (Fig. 3). We discuss this issue in lines 91-94 of the original manuscript. However, we further modified this paragraph to improve its clarity (lines 115 – 120).

fig 3: The yellow ocean warming colouring is problematic as the strongest indicated warming occurs that the end of HE1 with blue colouring.

We agree and improved all figures now showing ocean warming prior to Heinrich Events and the Younger Dryas in yellow and ice-sheet responses in grey.

94 which are related to two different ice-stream advances18.

Fact or inference?

This is the interpretation of Hodell et al. (2017) explaining the double peak in IRD by two different stages of ice-stream advances and IRD deposition during HE1. We revised the paragraph to make clear that we follow this interpretation (lines 115 – 120).

95-96 Overall, these findings unequivocally exhibit the close relationship between subsurface ocean temperature increases and the beginning of Heinrich Events in the North Atlantic over the last 30,000 years (Fig. 3).

"unequivocally" is an overstatement given the issues I've raised above

We re-phrased the paragraph to avoid overstatements (lines 120 – 123).

111 further crossing a critical threshold and destabilized the British Ice Sheet21.

Fact or inference?

According to cited work of Peck et al. (2008) the simultaneous appearance of anomalous warm temperatures and IRD deposition hints to destabilization of the proximal BIS. However, we improved this paragraph considering the work of Clark et al. (2012), that provides strong support to the idea of enhanced marine ice-margin retreat of the British-Irish Ice Sheet between ~19 to 17 ka BP (lines 142 – 145).

115-116 The quasi-simultaneous rise of subSST in both margins of the subpolar North Atlantic implying a basin-wide built-up of ocean heat that began at ~18.5 ka BP (Fig. 4b).

results from 2 cores "imply" basin-wide build-up????

We drew this conclusion based on three cores (including the study of Marcott et al., 2011). In fact, available records do support a simultaneous, basin-wide build-up of subsurface heat. Still, we toned down this statement taking into consideration the sparse availability of subSST records (lines 145 – 148).

117-118 This is largely consistent with a basin-wide increase of North Atlantic bottom water temperatures observed prior to Heinrich Event 1 22

240-241 22 Marcott, S. A. et al. Ice-shelf collapse from subsurface warming as a trigger for Heinrich events. Proc. Natl Acad. Sci. 108, 13415–13419 (2011).

"observed" or inferred?

The basin-wide increase of bottom water temperatures has been inferred from benthic Mg/Ca data (one site from the mid-depth western North Atlantic), further evaluated with a climate-model simulation that reveal basin-wide subsurface warming (Marcott et al. 2011) (lines 148 – 151).

119 We therefore propose that an enormous volume of heat was stored in the ocean-interior of the subpolar North Atlantic at this time.

another causal issue that is avoided. Even if the above claim is true, how do you get the heat up into the Northern part of the Labrador sea to reach Hudson Strait if this were the instability region (or is the instability region Gulf of St. Lawrence or ??? The boundary current for the Labrador Sea is largely fed by the Eastern Greenland current, at least presently.

We argue that IRD in core GeoB18530-1 is not dominated by terrigenous sediments from Gulf of St. Lawrence (please see comments above).

We do not suggest warming of sea surface temperatures in the Labrador Sea related to EGC or Labrador Current. The opposite is actually the case. We showed in Fig. 4e that sea surface temperatures in the North Atlantic do not show any sign of warming prior to Heinrich Event 1 different to the warming trends observed in the subSSTs in the subpolar western North Atlantic (Fig. 4b). We argue from subsurface temperatures (near ~ 150 m water depth; see also Methods and Extended Data Fig. 2 and 3) that the interior-ocean warmed prior to Heinrich Events close to Hudson Strait in the subpolar western North Atlantic region (please see also the revised Fig. 1b).

121-123 Third, proxy records of AMOC variability^{9,23} show that the deep ocean circulation weakened at the beginning of Heinrich Stadial 2 and 1, preceding Heinrich Events by several hundreds of years (Fig. 4d).

without showing age uncertainties, how is one supposed to evaluate this statement?

This inference comes from several studies showing that AMOC is in decline prior to Heinrich Event 1 and 2 (e.g. McManus et al., 2004, Lippold et al., 2009). In general, every paleoclimate record is affected by age uncertainties. However, we consider these studies as robust and see no reason to suspect the given conclusions. In particular, AMOC decline preceded Heinrich Events by 1 – 2 kyr and seems to be quite robust (as also discussed by Marcott et al., 2011).

To better assess the influence of age uncertainties on our subSST record (fig.4) we now include age uncertainties into the error margin (also requested by reviewer #3). This will allow the reader to evaluate the temporal relationship between increased subSSTs and subsequent IRD deposition during AMOC weakening (revised fig. 4b). We further improved the discussion on AMOC variability and the build-up of ocean heat in the western subpolar North Atlantic (lines 153 -162).

128-130 fact that the Younger Dryas is not characterized by a typical Heinrich Event with only and non-linear freshwater forcing delivered to the ocean via either the St. Lawrence River²⁴ and the Mississippi River²⁵ or the Mackenzie River²⁶. Nonetheless,

#The 1975 vintage reference ²⁵ would not have had AMS C14 dating. More recent records show clear shutdown of Mississippi discharge before the start of the Younger Dryas.

We now refer to a study by Carlson et al. (2007) suggesting freshwater discharge mainly from the St. Lawrence River (instead of the Mississippi or Mackenzie Rivers) during the Younger Dryas cold event (lines 163 – 167).

137-140 The rapid North Atlantic interior-ocean warming during AMOC weakening is a feedback mechanism also observed in several modeling simulations²⁷⁻²⁹. As soon as the AMOC is suppressed, due to freshwater hosing, immediate cooling and isolation of the surface ocean from subsurface waters is observed in the models ²⁹

#Why is only 29 cited in the second sentence if the claim in the first sentence is for all 3 27-29 studies?

We agree and changed the citation accordingly.

141-142 Physically, without cooling by convection, the ocean-interior warms, in particular in the equatorial- and South Atlantic³⁰.

Should make clear what heat source you are talking about. Ocean interiors don't just warm on their own.

The term convection describes the formation of cold and dense NADW in the North Atlantic, the deep branch of the AMOC. Thus, what the cited numerical models tell us is that the North Atlantic interior ocean would warm (by 1 – 2°C) simply from a weakened southward transport of NADW leading to a redistribution of heat in the Atlantic Basin (Rühlemann et al., 2004). Numerical models also suggest that, upon warming from reduced deep convection, enhanced interior-ocean transport of warm (and salty) subtropical water masses to the subpolar gyre system would dominate (advection) the northward heat transport to the subpolar Atlantic (e.g. He et al., 2020). Together with reduced downward mixing (due to isolation of the cold and fresh surface ocean from subsurface ocean in the North Atlantic and reduced NADW formation) the warm waters would be “trapped” in the subsurface ocean of the subpolar Atlantic (He et al., 2020). We revised the whole paragraph to clarify this issue (lines 169 – 191).

151-152 Moreover, as soon as the AMOC recovers from freshwater forcing, the surface ocean becomes warmer and subsurface ocean temperatures cooler due to active convection

convection where? Deep convection occurs in the high latitudes, still really confused. Do mean AMOC? THC? or?

We revised the whole paragraph to clarify these issues (also refer to our responses on the same topic above).

155-157 Our proxy time-series now provide solid evidence for the built-up of an ocean heat reservoir in the North Atlantic that occurred during AMOC slowdowns over the last 30,000 years

Over what exact geographic region? Would help interpret the claims presented.

We now better defined the region in which we observe subsurface warming in the North Atlantic (lines 188 – 191).

159 Several modeling studies discussed the substantial role of

What kind of "modelling studies"?

Here we referred to results based on ice-sheet modelling studies. We now included this information in the text (line 194).

166-170 Another major point is the water depth for the terminus of ice-shelves to be approximately 300 m for the LIS, analogue to modern Greenland, Svalbard and Alaskan marine-terminating-glaciers³³ We now provide the first indication yet for the proposed ocean-interior warming near the critical depth to initiate basal melting and further destabilization of the LIS, culminating into Heinrich Events.

#Water temperature at depth of terminus of ice shelf is not going to have much relevance for melt under an ice shelf. Water temperature at the deeper grounding line relative to upper level temperature is much more critical for driving sub-shelf melt, cf bouyant plume modeling papers of sub-shelf melt.

We agree that many different aspects are relevant to understand basal melting of ice-shelves (as discussed by Lazeroms et al., 2018), namely:: (i) the interplay of geometry of the ice-shelf base; (ii)

ocean temperature; (iii) parametrizations related to buoyant plume meltwater; and (iv) the need for detailed ocean–shelf–cavity models. We see the point that many relevant parametrizations (e.g., basal melting of an ice-shelf) are often insufficiently covered by paleo ice-sheet model experiments. However, we have to rely on state-of-the-art paleo ice-sheet and ocean model reconstructions telling us that subsurface ocean warming would be critical for melting of ice-shelves. In this context, we refer to Bassis et al. (2017) showing that, without an ice-shelf, relatively small fluctuations in subsurface ocean temperature would be sufficient to trigger Heinrich Events.

We revised the paragraph for a clearer discussion about what paleo ice-sheet models tell us on the role of subsurface ocean warming on ice-shelves and Heinrich Events under glacial conditions (lines 193 – 205).

185 Our study reveal a fundamental relationship between AMOC strength and ocean heat content in the North Atlantic

overstatement

We revised the paragraph and avoid overstatements (lines 186 – 188).

186-187 Our findings imply that a near future decline in AMOC will cause rapid increases of the North Atlantic heat content with unparalleled magnitude compared to instrumental time-series

overstatement

We revised the final paragraph and avoid overstatements (lines 220 – 223).

404-405 Seawater approximating paleo-salinity ($\delta^{18}\text{O}_{\text{sw-ivf}}$) We calculated the regional ice volume free $\delta^{18}\text{O}_{\text{sw}}$ record ($\delta^{18}\text{O}_{\text{ivf-sw}}$) consider..

so relying on single inversions with significant uncertainties that are ignored.

We want to point out that our interpretation and main conclusions are not relying on $\delta^{18}\text{O}_{\text{ivf-sw}}$ alone. However, we agree that uncertainty is in general quite large for $\delta^{18}\text{O}_{\text{ivf-sw}}$ reconstructions. We revised Fig. 3 including error estimates for $\delta^{18}\text{O}_{\text{ivf-sw}}$ (we missed before), even though the observed trends are still significant and support our conclusions.

References:

- Bond, G. C. et al. Evidence for massive discharges of icebergs into the North Atlantic Ocean during the last glacial period. *Nature* **360**, 245-249 (1992).
- Channell, J.E. et al. A 750-kyr detrital-layer stratigraphy for the North Atlantic (IODP sites U1302–U1303, Orphan Knoll, Labrador Sea). *Earth Planet. Sci. Lett.* **317**, 218–230 (2012).
- Hemming SR. Heinrich events: Massive late Pleistocene detritus layers of the North Atlantic and their global climate imprint. *Reviews of Geophysics* **42**(1): 1005 (2004).
- Hodell, D. A., Channell, J. E.T., Curtis, J. H., Romero, O. E. & Röhl, U. Onset of “Hudson Strait” Heinrich events in the eastern North Atlantic at the end of the middle Pleistocene transition (~640 ka)? *Paleoceanography* **23**, PA4218, doi:10.1029/2008PA001591 (2008).
- Hodell, D. A. et al. Anatomy of Heinrich Layer 1 and its role in the last deglaciation. *Paleoceanography* **32**, 284-303 (2017).
- Lazeroms, W.J.M. et al. Modelling present-day basal melt rates for Antarctic ice shelves using a parametrization of buoyant meltwater plumes. *The Cryosphere*, **12**, 49–70 (2018).
- Leng et al. Sedimentary and rock magnetic signatures and event scenarios of deglacial outburst floods from the Laurentian Channel Ice Stream. *Quat. Sci. Rev.* **186**, 27-46 (2018).
- Mulitza, S. et al. T. RV MARIA S. MERIAN, Cruise 39, St. John’s – St. John’s, 07.06. – 25.06.2014, MARIA S. MERIAN-Berichte, MSM39. *DFG-Senatskommission für Ozeanographie*, p. 89, https://doi.org/10.2312/cr_msm39 (2015).
- Nürnberg, D. et al. Western boundary current in relation to Atlantic Subtropical Gyre dynamics during abrupt glacial climate fluctuations. *Global and Planetary Change*, Volume **201** (2021)

Response to comments by Reviewer #2:

We thank Reviewer #2 for the helpful comment.

Summary:

This manuscript describes collected proxies of subsurface temperature and IRD of the last 30 kyrs in a sediment core located in the southwest subpolar North Atlantic Ocean. These proxies show a consistent pattern of subsurface warming prior to the IRD arrival during Heinrich events and the Younger Dryas. Because this is done on the same core, there can not be doubts about the relative timing of the shown variables.

The analysis of the time series argues in favor of the interpretation of Heinrich events being triggered through an induced destabilization of the Laurentide ice sheet by warming of the ocean in contact with its basal layer.

The current piece of work, together with Barker et al. 2015 nicely contribute, in my opinion, to the ultimate demonstration that Heinrich events were triggered through oceanic changes. And not the opposite, as it has been thought by a large part of the community for many years.

The manuscript is very well written, it reads fluently and the figures clearly support the description and interpretation of the presented proxies.

I recommend publication, essentially “as it is”.

Comments and suggestions:

I only have one minor suggestion regarding the discussed implications of their work for the current climate (at the end of the abstract and conclusions):

It is true that the AMOC is likely to be already suffering a strong decrease in its intensity (as shown by several papers), and that, as you show here, this has implications for the inner ocean heat reservoir. That being said, it is also known that the Greenland ice sheet is relatively “disconnected” to the ocean. The main factor for its decreasing mass in the coming centuries will be an increasingly negative surface mass balance. Although the warming waters of the fiords having a glacier terminy will indeed facilitate the grounding line migration and therefore enhance ice discharge to the ocean, this process will not be the dominant one for Greenland.

Therefore, I propose that the authors lower the supposed importance of their findings for understanding the future of the Greenland ice sheet announced in the abstract and at the end of the manuscript.

This can be done directly by rephrasing those parts of the manuscript, or it could also be done by expanding that line of thoughts regarding the future climate to the implications for Antarctica.

It is clear that the stability of the Antarctic ice sheet is directly linked to the Southern Ocean behavior during next centuries (see for example Golledge et al. 2015, Nature). What it is not so clear is the implications of a decreasing or halted AMOC on the Southern Ocean subsurface temperatures and thus the final implications on Antarctica.

If the authors want to keep their interpretation of the implications that their work has on future sea level rise, I recommend them to expand on the aforementioned issue.

We agree with the Reviewer and made changes to the future implications related to our findings. We followed the Reviewer suggestion and now discuss possible implications from future warming of the subsurface ocean in terms of modern, marine-terminating Arctic glaciers and the freshwater budget of the North Atlantic. Accordingly, we revised the respective paragraphs at the end of the abstract and at the end of the manuscript (lines 26 – 30; 220 – 223).

However, we decided not extending our implications to the Antarctic Ice Sheet. The reason is that we have no additional proxy-data from the Southern Ocean to start a discussion of subsurface ocean warming on the Antarctic Ice Sheet.

Response to comments by Reviewer #3:

We thank Reviewer #3 for the comprehensive and constructive comments.

Summary:

The authors present high resolution proxy data for a core from the subpolar N Atlantic. The methods employed are generally “traditional”. A big advantage of the current study is the high-resolution records from a very nice core. The data allow them to resolve timing of various tracers associated with Heinrich events over the past ~30 ka. Possible links between subsurface warming and icesheet instabilities are discussed.

To get the data published in Nature Communications, more work is needed. I have listed my comments in details below, and I hope they are useful for the authors to revise their manuscript.

Writing requires substantial improving.

Comments and suggestions:

1. age model and timing of events

Timing is very critical for the conclusion of this paper. Although the relative timing of Mg/Ca and Ca/Si is well defined (Fig. 2), it is necessary to know when changes in Mg/Ca & Ca/Se occurred in the context of N Atlantic climatic events (Fig. 4a). That is, a robust calendar age model is needed.

Currently, the age model is constructed using model-based surface reservoir ages. I don't think this is a good/robust practice. The problematic age model is reflected by asynchronous warming at SU8118 (and possibly MD01-2461) and NGRIP at Bolling onset and YD termination and possibly at HS2 termination (Fig. 4). For example, how can warming at SU8118 lag by so long time relative to NGRIP d18O at Bolling onset and YD termination? Unless the authors have a sound interpretation for such warming offsets in timing, the age model needs to be reconsidered.

We thank reviewer #3 for for detailed inspection of fig. 4. We carefully checked the time-series of MD01-2461 and SU8118 and found, unfortunately, two small bugs in published time-series given in Figure 4. We note that none of these issues afflicting our main conclusions. The first bug is visible in MD01-2461 during the Bolling/Allerod. In this case we found one AMS14C age was simply not transferred correctly to our dataset. This issue has been fixed (fig. 4d in revised manuscript). The second bug results in an apparent offset between NGRIP d18O and SSTs from SU8118. The reason for the offset is that we considered published planktonic foraminiferal AMS 14C ages from SU8118 as “raw ages” (given in Bard et al., 1987). Unfortunately, we overlooked information that presented ages have been corrected for a “mean present apparent age of surface water”. Thus, the offset (that is too young ages in original figure 4) comes from adding reservoir age calculations to AMS14C ages that have been “corrected” before. We now fixed this issue resulting in quite more reasonable SST development in the North Atlantic (fig. 4e in revised manuscript).

However, we do not agree with the opinion of the reviewer that our age model construction for core GeoB18530-1 is not robust/good practice for several reasons. First, the age model is based on 20 AMS 14C ages (age control points) without any tuning to other records (e.g. NGRIP). This is important because it allows discussing all presented proxy records (independently) to Heinrich Events deposited in the same core excluding any circular reasoning (from tuning). Second, we considered most recent simulations of surface-reservoir ages from Butzin et al. (2020) to correct our marine radiocarbon ages for variable surface water reservoir ages through time. We feel this approach gives at least a reasonable approximation in temporal variability in past surface reservoir ages, whose are largely

unknown. Third, our age model is based on an ensemble of 2,000 age-depth realizations allowing us to realistically assess the influence of the temporal error on the subSST time series.

We now added statistics including mean age of our age model ensemble along with 95% confidence interval to figure 4.

I suggest trying Nps% to obtain an independent age model for the core. Nps% could be useful to pin down timing for YD, HS2, and Bolling onset (see Thornalley et al., 2011).

Unfortunately, there is no Nps% record available for core GeoB18530-1. Nevertheless, we pinned down transitions to HE2 and HE1 by AMS14C ages in GeoB18530-1 (Fig.3 and 4). Moreover, the timing of Heinrich Events is consistent to other studies dealing with Heinrich Events in the North Atlantic. For example, the onset of Heinrich Event 1 at 17.07 ka BP in core GeoB18530-1 is very close to the reported age of 17.1 ka BP of Hodell et al. (2017) (lines 103 - 106).

It is necessary to know when warming and Ca/Sr changes occurred, including starting and ending dates. For example, was warming started during a certain stadial or at the end of the preceding interstadial?

Based on our age model we observe an increase to warmer subSST during HS2, HS1 and the Younger Dryas (fig. 4 in revised manuscript). We now provide mean ages from our age model ensemble for onset of warming events in the revised manuscript (lines 156 – 160).

A robust age model can be valuable to reveal possible new mechanisms controlling stadial icesheet instabilities.

The age model is based on 20 AMS 14C ages (age control points) without any tuning to other records (e.g. NGRIP). This is important because it allows discussing all presented proxy records (independently) to Heinrich Events deposited in the same core as well as other records excluding any circular reasoning (from tuning). Second, we considered most recent simulations of surface-reservoir ages from Butzin et al. (2020) to correct our marine radiocarbon ages for variable surface water reservoir ages through time. We feel this approach gives at least a reasonable approximation in temporal variability in past surface reservoir ages, whose are largely unknown. Third, our age model is based on an ensemble of 2,000 age-depth realizations allowing us to realistically assess the influence of the temporal error on the subSST time series. Thus, we consider our age model as robust.

2. Icesheet instability indicators

The use of Ca/Sr as a tracer to reflect icesheet instability is implicit, but needs to be explicitly spelled out in the main text.

Ice-sheet instability is inferred from extremely well-preserved Heinrich Layers in core GeoB18530-1 (fig. 2 a; Mulitza et al., 2015). These layers show typically low $\delta^{18}O$ values as expected from freshwater release from iceberg discharge during Heinrich Events (green curve, fig. 3c) Moreover, the onset of IRD deposition is well constrained by AMS14C ages allowing us to further define the onset Heinrich Event 1 and Heinrich Event 2.

Principally, the Ca/Sr signal serves as a proxy for the geochemical signature and provenance of existing IRD layers. We found that elevated Ca/Sr ratios of existing IRD layers in core GeoB18530-1 matches the common signature of detrital carbonate associated with Hudson Strait Heinrich Events (e.g. Hodell et al., 2008; Channel et al., 2012 or Hodell et al., 2017). We revised the paragraph to avoid confusion and pointing out more clearly the use of Ca/Sr as indicator for detrital carbonate coming from Hudson Strait Heinrich Events (lines 92 –95).

Also, does it mean that icesheet is stable if Ca/Sr values remain low?

This is hard to tell. The Ca/Sr signal is telling us something about the provenance of IRD Layers preserved in core GeoB18530-1. In case of GeoB18530-1 the elevated Ca/Sr ratios point to Hudson Strait/Hudson Bay as source of IRD. It is possible the Laurentian Ice Sheet experienced more transient conditions with discrete melting and freshwater release several times during e.g. MIS2, however without deposition of prominent IRD Layers.

Nevertheless, in our manuscript we focus on catastrophic ice sheet surges during Heinrich Events. To better acknowledge the limitations of our study we changed the title to “Subsurface ocean warming preceded abrupt ice-sheet instabilities during Heinrich Events”.

Additionally, some secondary Ca/Sr peaks are seen (Fig. 2c; 430 cm, 150 cm, etc); do they also reflect icesheet instabilities? What is the threshold to be used to reflect an icesheet instability? Please define a value in Ca/Sr. This seems arbitrary as it stands.

The secondary peaks at 430cm and 150cm are paralleled by slightly warmer subSSTs and could reflect unknown, more discrete ice-sheet instability events without deposition of IRD layers in core GeoB18530-1. The Ca/Sr signal serves as a proxy for provenance of existing IRD layers. We have no indication for IRD deposition (no IRD layers) at site GeoB18530-1 during secondary Ca/Sr peaks. In principle, the lack of IRD layers does not exclude smaller (unknown) ice-sheet instabilities related to subsurface warming. However, we focus on Heinrich Events that are clearly associated with deposition of Heinrich Layers at site GeoB18530-1. To better acknowledge the limitations of our study we changed the title to “Subsurface ocean warming preceded abrupt ice-sheet instabilities during Heinrich Events”.

3. Mechanisms for the early warming and icesheet stability

What’s new that can be learnt from the data presented? Is there any new mechanism for the icesheet instability, and for the warming prior to the icesheet instabilities?

My impression is that the new data provide evidence to support subsurface warming that may cause icesheet melting during Heinrich stadials. The related ideas have long been published (e.g., ref 22). To get this work published in Nature Communications, I encourage the authors to think deeply about new findings that can advance our understanding of icesheet stability and abrupt climate changes in the North Atlantic (and broader) regions.

We agree with the reviewer that our new data provide robust evidence in support of subsurface warming as potential trigger for ice-sheet instabilities during Heinrich Events. It is also true that these ideas have long been published (e.g. Shaffer et al., 2004). We also note that most studies discussing subsurface warming as trigger for Heinrich Events are based on model experiments but lacking any evidence from proxy data. The work of Marcott and colleagues made a first valuable step using benthic Mg/Ca temperatures from a mid-depth core in the North Atlantic to show bottom water temperatures are rising before some Heinrich Events. Nevertheless, if we believe the model simulations (in favor of subsurface warming as trigger for ice-sheet instabilities) bottom water temperatures are hardly relevant for melting of an ice-shelf or ice-sheet instability, even if thoroughly discussed in the paper of Marcott et al. (2011).

According to the models, more relevant for ice-shelf disintegration and ice-sheet instabilities is subsurface warming of the upper few hundreds of meters of the interior-ocean to initiate melting of the ice shelf and to trigger Heinrich Events (e.g. Álvarez-Solas et al., 2010; 2011). In more detail, the build-up of ocean heat near the grounding line of ice-shelves is the critical value in these models to trigger rapid retreat of the ice-margin around the Labrador Sea and Heinrich Events. However, there is no subSST proxy record available so far, impeding evaluation of subsurface ocean warming as trigger of Heinrich Events. We now provide first robust evidence for repeated and massive subsurface warming near Labrador Sea. Thus, our proxy time-series confirms theoretical model experiments and provide first robust evidence for interior-ocean warming of the western subpolar North Atlantic as trigger for Heinrich Events. We also verify and expand results from theoretical model experiments proposing interior-ocean warming of the western subpolar North Atlantic as natural feedback mechanism to slowdowns of the AMOC during the past 27,000 years. Finally, our study lend strong support for an ocean forcing of abrupt ice-sheet instabilities during Heinrich Events. We have no doubt these findings are new and highly relevant for a broader audience dealing with past and future abrupt climate change associated with ice-sheet instabilities.

Some large increases in Mg/Ca are observed at other times such as 420 cm, 200 cm, etc (Fig. 2); what caused these warming events? Also, these warming events do not follow large increases in

Ca/Sr (that is, icesheet instability). For HS2, we see double warming prior to the large Ca/Sr peak. In contrast to the warming around 25.5 ka which was followed by large Ca/Sr, warming at ~26.5 ka (of similar magnitude as occurred at 25.5 ka) did not “initiate” any Ca/Sr rise. Do these observations suggest that warming does not always initiate instability of icesheets? If so, implications for future climate would be ambiguous, and revision is necessary (e.g., Abstract and the last paragraph).

We can only speculate about the secondary subsurface warming events at 420cm and 200cm. Given the apparent limitations in resolution of Pa/Th we are not able to align smaller (unknown), more modest warmings to changes in AMOC. As pointed out before, ice-sheet instabilities are possible without IRD deposition at site GeoB18530-1. This may be the case for the small warming event at 26.6 ka BP we see modest freshening (fig. 3c) but no IRD layer. Another example is the Younger Dryas Event, where we have subsurface warming and freshening but a delay in IRD deposition at site GeoB18530-1. In case of Younger Dryas, it has been proposed that freshwater was mainly delivered from Gulf of St. Lawrence (discussed in lines 162 –167).

To better acknowledge the limitations of our study, we changed the title of our study to “Subsurface-ocean warming preceded abrupt ice-sheet instabilities during Heinrich Events”.

4. Data processing and proxy interpretation

Statistical analyses – Monte Carlo simulation of reconstructions are necessary for records shown in Fig. 4, especially for records generated in this study.

We now added statistics including mean age of our age model ensemble along with 95% confidence interval for the subSST record to figure 4.

Additionally, what is the point of showing $\delta^{18}\text{O}_{\text{ivc-sw}}$? The data appear to be not well used.

In our case, the $\delta^{18}\text{O}_{\text{ivc-sw}}$ nicely reflects the expected increase in salinity during warming of subsurface waters. In general, the reconstruction of salinity from $\delta^{18}\text{O}_{\text{ivc-sw}}$ is afflicted by large uncertainties (also claimed by Reviewer #1) and thus we toned down interpretation of salinity variations.

Detailed comments:

Abstract: unless the authors can provide model results for future scenarios, implications for the future should be brief and limited to one sentence.

We agree. We rephrased the implications of our study following the suggestions from Reviewer #2 (lines 26 – 30).

Main: It starts very abruptly into data/methods. I strongly suggest adding one opening paragraph to describe the current status of related studies and why this study is warranted.

We agree and added a paragraph providing a brief discussion on proposed causes of Heinrich Events and what is needed to further evaluate proposed mechanisms (lines 32 - 58).

Line 60-62: correct grammar.

Fixed.

Line 64-65: how to define the timing mentioned here? When/where is the onset of HEs? See my concerns above about using model-based surface reservoir ages to convert 14C to calendar ages. Is YD treated as a HE? If so, where is the onset of YD? can the small Ca/Sr peak at ~335 cm be treated as the start of IRD rise? If so, why cannot other similarly small Ca/Sr rises (e.g., 220 cm, 440 cm, etc) be treated as IRD rises (and icesheet instabilities)? A value of Ca/Sr needs to be defined to detect IRD events.

The beginning of Heinrich Events is well defined (and visible) by respective IRD (Heinrich) Layers and simultaneous anomalies in the $\delta^{18}\text{O}$ -signal in core GeoB18530-1 (we refer to detailed discussion above). The onset of Heinrich Layers is pinned down by AMS14C ages. For example we got a calibrated median age of 17.07 ka BP for the beginning of Heinrich Event 1 (lines 103 - 106) very close to the reported age of 17.1 ka BP for the beginning of IRD deposition during Heinrich Event 1 (Hodell et al., 2017).

The small peak at the beginning of the Younger Dryas is related to a dropstone (Mulitza et al., 2015) and we added this information to figure 2. The secondary peaks at 220 cm and 420 cm are not accompanied by IRD layers in core GeoB18530-1.

The Ca/Sr signal tells us that the source of IRD is the area around Hudson Strait/Hudson Bay. It is also possible that smaller events points to the instability of the LIS but without significant IRD deposition and are related to e.g. freshwater coming e.g. Gulf of St. Lawrence (as proposed by Reviewer #1 for the Younger Dryas, discussed in lines 163 –167). To acknowledge these limitations we changed the title of our manuscript to “Subsurface ocean warming preceded abrupt ice-sheet instabilities during Heinrich Events”.

Line 87-90: based on Fig. 4c, we do see some warming coeval with subSST rise (Fig. 4b) around HE1 (17.5 ka) and YD, inconsistent with descriptions here.

We can't see any warming in Fig. 4c (SSTs) during HE1 or Younger Dryas? However, as discussed above the presented age model of SU8118 suffered from small bugs in the previous manuscript, which we fixed in the revised manuscript (fig. 4e). The opposing trends between subSST and SSTs are now even more pronounced (in particular during HS2, HS1 and the Younger Dryas).

Paragraph starting from line 95: description unclear; substantial improvement is necessary.

We revised the paragraph.

Line ~116: if one looks into Figs. 3 and 4 (which is difficult), the Ca/Sr peak labelled as “YD” in Fig. 3 appears to correspond to a time after YD shown in Fig. 4 (corresponds to PB?). I suggest adding Ca/Sr in Fig. 4 to facilitate comparing of relative timing of various records. Currently, it is very difficult, if not possible, to read two figures back and forth.

We agree. We are sorry for the confusion and now revised the figures 3 and 4 adding the Ca/Sr record to figure 4.

Line 121: “a fundamental relationship” – what is it? Specify.

We revised the whole paragraph to better specify the temporal relationship between AMOC and subSSTs in the western North Atlantic (lines 153 – 167).

Line 140-142: reference? Note that CESM has long simulation runs.

We agree. These are specific simulations covering e.g. the mid Pliocene, last interglacial or mid Holocene (e.g. Feng et al., 2020). At the end, we decided to remove the statement from the paragraph to avoid too much confusion to the reader.

Line 82-84: “close relationship”, please specific what the relationship is.

At this point, we argue for a close temporal relationship of warmest subSSTs and the onset of IRD deposition in core GeoB18530-1. We revised our statement to make this point clear (revised lines 120 – 123).

Line 157: “paramount role”, “fundamental conclusion”, so far, similar wording appears several times. Please demonstrate the role and what is new in terms mechanisms that gleaned from the new records! Otherwise, avoid exaggerating the claim.

Line 171: again “fundamental relationship”... explain- what is it?

We revised major parts of the manuscript in terms of better describing the role of subSST on ice-sheet instability during Heinrich Events. In general, we avoid wording that do exaggerating the claim. The whole manuscript now appears more descriptive.

Line 328-335: Al/Ca detection limit? Add unit (mmol/mol) after Al/Ca and Mg/Ca values wherever necessary.

We added this information into Methods (lines 377 – 378). We fixed units to respective values in the text (lines 383 – 396).

Fig. 4: add Ca/Sr in the figure to facilitate comparisons; add yellow bars to indicate warming prior to ice-sheet instabilities (Ca/Sr). Add age control points in the figure as well. For HS2, AMOC recover led NGRIP warming – how cause this large leading, or is it an age model issue?

We agree and revised fig. 4 following the recommendations of the reviewer. The Pa/Th time-series >19 ka BP comes from the study of Lippold et al. (2009). We found that the age model for used core (ODP1063) is based on correlation to another core GPC 5 and not confirmed by age control points (AMS14C ages) at Site 1063. It is thus possible that the leading of AMOC is an age model issue (seems to be the case for early AMOC slowdown prior to HS2; Fig. 4f). Moreover, resolution of Pa/Th is often low (unfortunately) near or close critical transitions e.g. at the beginning and end of Heinrich Stadial 2 or YD, adding further uncertainty to the time-series.

Thornalley, D. J. R., Barker, S., Broecker, W., Elderfield, H. & McCave, I. N. *The Deglacial Evolution of North Atlantic Deep Convection. Science 331, 202-205 (2011).*

References:

- Bard, E. et al. Retreat velocity of the North Atlantic polar front during the last deglaciation determined by ¹⁴C accelerator mass spectrometry. *Nature*, Vol 328 (27) (1987).
- Feng et al. Increased Climate Response and Earth System Sensitivity From CCSM4 to CESM2 in Mid-Pliocene Simulations. *Journal of Advances in Modeling Earth Systems*, 12, e2019MS002033 (2020)

Reviewers' Comments:

Reviewer #1:

Remarks to the Author:

The revised version is much improved and addresses most of my concerns raised. However, a few remain that are easily addressed as indicated below (and should not require re-review). Once addressed, I would see this in good form for publication in Nature Communication, providing some significant evidence in favour of sub-surface ocean warming triggering at least 2 of the past 6 Heinrich events.

Lev Tarasov

(NOTE '#' denotes my first review comments, '##' denotes my new review comments, '#a' denotes author response)

title:

The authors have implicitly rejected certain hypotheses about Heinrich events in their title and opening paragraph. To my knowledge, it has yet to be ruled out that Heinrich events are not due to a collapse of an ice shelf at the mouth of Hudson Strait (Hulbe et al, 2004, cited ref # 8 in the submission) as opposed to an onset of Hudson Strait streaming. This alternate hypothesis may or may not involve "ice-sheet instabilities". Though to my judgement the balance of evidence favours the ice-stream onset hypothesis (perhaps triggered by ice shelf collapse), I see no basis for the implicit treatment as a scientific fact. It could be that the authors consider "ice-sheet instabilities" to include ice shelf collapse, but that is not how I suspect many in the field would interpret the statement. The remedy for the title is simple: just remove "abrupt ice-sheet instabilities during". Otherwise clarify early on that "ice-sheet instabilities" potentially includes "ice-shelf instabilities".

I also find the title problematic since the presented record only spans two of 6 Heinrich events during the last glacial cycle. This issue can easily be remedied by insertion of "during MIS II". These results provide the first solid evidence for the massive accumulation of 20 ocean heat near the critical depth to trigger melting of marine-terminating

We studied fluctuations in subSST and salinity relative to the occurrence of Heinrich

The text current lacks any concrete statement of what depth range "subSST" is, making it hard to interpret the validity of various claims made. Are you talking just about the mixed layer? Some set depth (100 m)? Or do you include the thermocline?

44 global ice volume-free oxygen isotopic composition of seawater
ice volume-free is confusing, how about "ice-volume removed"

#a Both ice-volume-free as well as ice-volume-corrected have been used in

similar studies but not "ice- volume-removed". We decided to stay with the terminology given by e.g. Nürnberg et al. (2021).

I ran this by my research group (paleo ice sheet and climate modellers), and all were confused by "ice-volume-free". They didn't know if this meant free of ice volume signal or free of ice volume corrections.

80 against Heinrich Events associated with LIS instabilities near Hudson Strait because ## "near" would to me preclude "within". This could preclude the binge-purge hypothesis. -> "near or within"

121-123 Third, proxy records of AMOC variability^{9,23} show that the deep ocean circulation weakened at the beginning of Heinrich Stadial 2 and 1, preceding Heinrich Events by several hundreds of years (Fig. 4d).

without showing age uncertainties, how is one supposed to evaluate # this statement?

#a This inference comes from several studies showing that AMOC is in decline prior to Heinrich Event 1 and 2 (e.g. McManus et al., 2004, Lippold et al., 2009). In general, every paleoclimate record is affected by age uncertainties. However, we consider these studies as robust and see no reason to suspect the given conclusions. In particular, AMOC decline preceded Heinrich Events by 1 – 2 kyr and seems to be quite robust (as also discussed by Marcott et al., 2011).

#a To better assess the influence of age uncertainties on our subSST record (fig.4) we now include age uncertainties into the error margin (also requested by reviewer #3). This will allow the reader to evaluate the temporal relationship between increased subSSTs and subsequent IRD deposition during AMOC weakening (revised fig. 4b). We further improved the discussion on AMOC variability and the build-up of ocean heat in the western subpolar North Atlantic (lines 153 -162).

this still doesn't address my concern. How is one supposed to evaluate the phasing of two paleo timeseries from different cores, when the age uncertainty of one of the records is not communicated? If the source publication doesn't provide an age uncertainty, then explicitly state what you think is a conservative estimate. Otherwise everybody will be making their own informed or ill-informed guesses and that does not help science progress.

Relatedly, for the caption of Fig 3, please indicate whether the age control triangles cover 1 or 2 sigma ranges.

128-130 fact that the Younger Dryas is not characterized by a typical Heinrich Event with only and non-linear freshwater forcing delivered to the ocean via either the St. Lawrence River²⁴ and the Mississippi River²⁵ or the Mackenzie River²⁶. Nonetheless,

#The 1975 vintage reference ²⁵ would not have had AMS C14 dating. More recent records show clear shutdown of Mississippi discharge before the start of the Younger Dryas.

#a We now refer to a study by Carlson et al. (2007) suggesting freshwater discharge mainly from the St. Lawrence River (instead of the Mississippi or Mackenzie Rivers) during the Younger Dryas cold event (lines 163 – 167).

The issue of Mackenzie versus St. Lawrence discharge contributions to Younger Dryas (YD) onset is far from settled. Carlson et al 2007 do not consider relevant physical oceanography issues (cf Tarasov and Peltier, Nature, 2005 and Condron and Winsor, PNAS, 2012). Furthermore, there are more recent geological papers pointing to significant Mackenzie discharge at YD onset (eg Murton et al, 2010).

Reviewer #3:

Remarks to the Author:

I have read through the revised version of the manuscript along with the rebuttal, and am satisfied with their responses. So, I recommend this work for publication.

Response to comments by Reviewer #1:

We thank Lev Tarasov for indispensable effort to improve this study.

Summary:

The revised version is much improved and addresses most of my concerns raised. However, a few remain that are easily addressed as indicated below (and should not require re-review). Once addressed, I would see this in good form for publication in Nature Communication, providing some significant evidence in favour of sub-surface ocean warming triggering at least 2 of the past 6 Heinrich events.

Comments and suggestions:

The authors have implicitly rejected certain hypotheses about Heinrich events in their title and opening paragraph. To my knowledge, it has yet to be ruled out that Heinrich events are not due to a collapse of an ice shelf at the mouth of Hudson Strait (Hulbe et al, 2004, cited ref # 8 in the submission) as opposed to an onset of Hudson Strait streaming. This alternate hypothesis may or may not involve "ice-sheet instabilities". Though to my judgement the balance of evidence favours the ice-stream onset hypothesis (perhaps triggered by ice shelf collapse), I see no basis for the implicit treatment as a scientific fact. It could be that the authors consider "ice-sheet instabilities" to include ice shelf collapse, but that is not how I suspect many in the field would interpret the statement. The remedy for the title is simple: just remove "abrupt ice-sheet instabilities during". Otherwise clarify early on that "ice-sheet instabilities" potentially includes "ice-shelf instabilities".

We agree and now avoiding "ice-sheet instabilities" in the title. The new title has been changed to: "Subsurface ocean warming preceded Heinrich Events".

I also find the title problematic since the presented record only spans two of 6 Heinrich events during the last glacial cycle. This issue can easily be remedied by insertion of "during MIS II". These results provide the first solid evidence for the massive accumulation of 20 ocean heat near the critical depth to trigger melting of marine-terminating

We are not in favor of adding "MIS2" to the title. The reason is that MIS2 is a very technical term and only a small fraction of the community would understand it. We also believe it is not feasible to convey all details in the title. However, we made a clear statement in the second sentence of the Abstract about the time-coverage of our study (line 8).

We studied fluctuations in subSST and salinity relative to the occurrence of Heinrich
The text current lacks any concrete statement of what depth range "subSST" is, making it hard to interpret the validity of various claims made. Are you talking just about the mixed layer?
Some set depth (100 m)? Or do you include the thermocline?

We now provide a statement in the Abstract and Introduction about the definition of subSST (lines 3 – 8; 52 - 55).

44 global ice volume-free oxygen isotopic composition of seawater
ice volume-free is confusing, how about "ice-volume removed"

#a Both ice-volume-free as well as ice-volume-corrected have been used in similar studies but not "ice-volume-removed". We decided to stay with the terminology given by e.g. Nürnberg et al. (2021).

I ran this by my research group (paleo ice sheet and climate modellers), and all were confused by "ice-volume-free". They didn't know if this meant free of ice volume signal or free of ice volume corrections.

We changed the term “ice-volume-free” to “ice-volume-corrected” throughout the manuscript to avoid confusions to the paleo ice-sheet and climate-modelers community.

**80 against Heinrich Events associated with LIS instabilities near Hudson Strait because
"near" would to me preclude "within". This could preclude the binge-purge hypothesis.
-> "near or within"**

Done.

21-123 Third, proxy records of AMOC variability^{9,23} show that the deep ocean circulation weakened at the beginning of Heinrich Stadial 2 and 1, preceding Heinrich Events by several hundreds of years (Fig. 4d).

**# without showing age uncertainties, how is one supposed to evaluate
this statement?**

#a This inference comes from several studies showing that AMOC is in decline prior to Heinrich Event 1 and 2 (e.g. McManus et al., 2004, Lippold et al., 2009). In general, every paleoclimate record is affected by age uncertainties. However, we consider these studies as robust and see no reason to suspect the given conclusions. In particular, AMOC decline preceded Heinrich Events by 1 – 2 kyr and seems to be quite robust (as also discussed by Marcott et al., 2011).

#a To better assess the influence of age uncertainties on our subSST record (fig.4) we now include age uncertainties into the error margin (also requested by reviewer #3). This will allow the reader to evaluate the temporal relationship between increased subSSTs and subsequent IRD deposition during AMOC weakening (revised fig. 4b). We further improved the discussion on AMOC variability and the build-up of ocean heat in the western subpolar North Atlantic (lines 153 -162). this still doesn't address my concern. How is one supposed to evaluate the phasing of two paleo timeseries from different cores, when the age uncertainty of one of the records is not communicated? If the source publication doesn't provide an age uncertainty, then explicitly state what you think is a conservative estimate. Otherwise everybody will be making their own informed or ill-informed guesses and that does not help science progress.

In fact, the source publication doesn't discuss possible age uncertainties from variable reservoir ages. Thus, we added a conservative estimate of age uncertainties, which is in the range of several hundreds of years for paleoceanographic time-series based on radiocarbon ages (lines 144 -146).

Relatedly, for the caption of Fig 3, please indicate whether the age control triangles cover 1 or 2 sigma ranges.

The triangles only inform about the position of age control points. The 1 and 2 sigma ranges are given in the Supplementary Table 1.

128-130 fact that the Younger Dryas is not characterized by a typical Heinrich Event with only and non-linear freshwater forcing delivered to the ocean via either the St. Lawrence River²⁴ and the Mississippi River²⁵ or the Mackenzie River²⁶. Nonetheless,

#The 1975 vintage reference ²⁵ would not have had AMS C14 dating. More

recent records show clear shutdown of Mississippi discharge before the start of the Younger Dryas.

#a We now refer to a study by Carlson et al. (2007) suggesting freshwater discharge mainly from the St. Lawrence River (instead of the Mississippi or Mackenzie Rivers) during the Younger Dryas cold event (lines 163 – 167).

The issue of Mackenzie versus St. Lawrence discharge contributions to Younger Dryas (YD) onset is far from settled. Carlson et al 2007 do not consider relevant physical oceanography issues (cf Tarasov and Peltier, Nature, 2005 and Condron and Winsor, PNAS, 2012). Furthermore, there are more recent geological papers pointing to significant Mackenzie discharge at YD onset (eg Murton et al, 2010).

We followed the suggestion from Reviewer #1 and now added the publication from Murton et al. (2010) to provide a more balanced discussion of Mackenzie versus St. Lawrence River discharge contributions at the onset of the Younger Dryas (lines 155 – 158).